# Nonlinear mechanics of human mitotic chromosomes

Anna E. C. Meijering[1,3], Kata Sarlós[2,3], Christian F. Nielsen[2,3], Hannes Witt[1], Janni Harju[1], Emma Kerklingh[1], Guus H. Haasnoot[1], Anna H. Bizard[2], Iddo Heller[1], Chase P. Broedersz[1✉], Ying Liu[2], Erwin J. G. Peterman[1,4], Ian D. Hickson[2,4✉] & Gijs J. L. Wuite[1,4✉]

In preparation for mitotic cell division, the nuclear DNA of human cells is compacted into individualized, X-shaped chromosomes[1]. This metamorphosis is driven mainly by the combined action of condensins and topoisomerase IIα (TOP2A)[2,3], and has been observed using microscopy for over a century. Nevertheless, very little is known about the structural organization of a mitotic chromosome. Here we introduce a workflow to interrogate the organization of human chromosomes based on optical trapping and manipulation. This allows high-resolution force measurements and fluorescence visualization of native metaphase chromosomes to be conducted under tightly controlled experimental conditions. We have used this method to extensively characterize chromosome mechanics and structure. Notably, we find that under increasing mechanical load, chromosomes exhibit nonlinear stiffening behaviour, distinct from that predicted by classical polymer models[4]. To explain this anomalous stiffening, we introduce a hierarchical worm-like chain model that describes the chromosome as a heterogeneous assembly of nonlinear worm-like chains. Moreover, through inducible degradation of TOP2A[5] specifically in mitosis, we provide evidence that TOP2A has a role in the preservation of chromosome compaction. The methods described here open the door to a wide array of investigations into the structure and dynamics of both normal and disease-associated chromosomes.

The structure of eukaryotic chromosomes changes markedly as cells traverse the cell division cycle. In interphase, nuclear DNA has a diffuse appearance and individual chromosomes are not discernible. As cells enter mitosis, the replicated chromosomes condense into compact, cylindrical structures comprising two sister-chromatid arms that mature into the iconic chromosome X shape in metaphase, in which the sister chromatids are held together only at the centromere. The sisters are then segregated to the nascent daughter cells in anaphase and telophase using force applied by the mitotic spindle[6,7]. The prevailing model of chromosome organization posits that consecutive loops of chromatin are organized in a helical staircase conformation[8] that is anchored to a central protein scaffold, with condensins I and II and TOP2A being key factors in mitotic chromosome formation[3,9–11]. Although condensins have also been shown to be essential for the maintenance of a compacted chromosome structure[2,12], there are conflicting views on the role of TOP2A in this process[5,13].

The mechanics and dynamics of many biomolecules have been elucidated by the use of micromechanical measurements, such as atomic force microscopy, and magnetic and optical tweezers[14]. Nevertheless, few mechanical studies have been performed on chromosomes[15]. A series of studies quantifying the mechanical stability of amphibian and human chromosomes, analysed by stretching chromosomes with micropipettes[16–18], revealed that they can be reversibly stretched by up to five times their native length by applying forces in the nanonewton range, and that depletion of condensins results in decreased chromosome stiffness[19]. Moreover, studies using Hi-C and super-resolution fluorescence microscopy[20,21] have provided important information on chromosome structure and organization. To gain direct access to the dynamic structural features of chromosomes, while avoiding fixation and ensemble averaging, we introduce here a workflow to analyse the mechanics and architecture of mitotic chromosomes using a combination of optical tweezers, fluorescence microscopy and microfluidics, which readily allows the manipulation of individual native chromosomes with nanometre precision and piconewton force resolution.

## Handling and visualizing chromosomes

To study native metaphase chromosomes using optical tweezers, we purified chromosomes with biotinylated telomeric ends, which served as molecular 'handles' for site-specific attachment to streptavidin-coated microspheres (Fig. 1a). Telomere-specific biotinylation was achieved through fusion of BirA protein to telomere repeat-binding factor 1 (TRF1)[22]. Biotin treatment of either U2OS-BirA-TRF1 cells or HCT116 cells transduced with a TRF1-BirA lentivirus resulted in biotin incorporation at approximately 98% of telomeres ($n_{tot}$ = 1,434; Extended Data Fig. 1a). We then optimized a chromosome isolation protocol[23] that

[1]Department of Physics and Astronomy and LaserLaB Amsterdam, Vrije Universiteit Amsterdam, Amsterdam, The Netherlands. [2]Center for Chromosome Stability and Center for Healthy Aging, Department of Cellular and Molecular Medicine, University of Copenhagen, Copenhagen, Denmark. [3]These authors contributed equally: Anna E. C. Meijering, Kata Sarlós, Christian F. Nielsen. [4]These authors jointly supervised this work: Erwin J. G. Peterman, Ian D. Hickson, Gijs J. L. Wuite. ✉e-mail: c.p.broedersz@vu.nl; iandh@sund.ku.dk; g.j.l.wuite@vu.nl

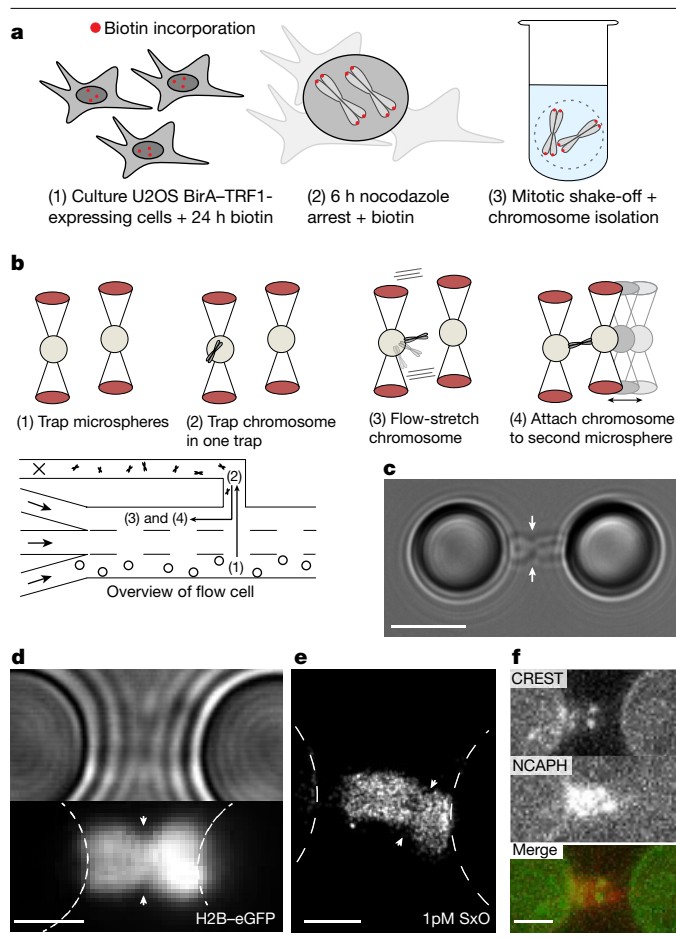

**Fig. 1 | Workflow for chromosome attachment and visualization. a**, Diagram depicting the experimental workflow. After addition of biotin to U2OS cells expressing BirA–TRF1, proteins located at telomeric ends are covalently biotinylated. Chromosomes were then purified from cells arrested in prometaphase by treatment with nocodazole. **b**, Schematic depiction of chromosome attachment to microspheres in a microfluidic flow cell with parallel channels (bottom left). After trapping two streptavidin-coated microspheres (1), a chromosome is attached to one microsphere by exploiting the attraction force that the optical trap exerts on the chromosome (2). The chromosome is then flow-stretched (3), bringing it into the imaging plane, and attached to the second microsphere (4). **c**, Representative bright-field image of a mitotic chromosome showing telomeric attachment of the four chromatid ends. The centromeric region is discernible as a constriction (arrows). Scale bar, 4 μm. **d**, Representative fluorescence image of H2B–eGFP (bottom) and corresponding bright-field image (top). Note that the chromosome in **d** was positioned to be in the focal plane for fluorescence imaging and not bright-field imaging. Scale bar, 2 μm. **e**, Representative BALM super-resolved image of SYTOX orange (SxO) binding events. The centromeric region is indicated with arrowheads. Scale bar, 1 μm. **f**, An example of immuno-staining of NCAPH and CREST to show the localization of condensin I along the chromatid scaffold and two foci that reveal the position of the centromeres, respectively. Scale bar, 2 μm.

yielded highly concentrated (around $10^6$ chromosomes per ml) native mitotic chromosomes devoid of cell debris and cytoskeletal contaminants (see Methods), which were suitable for telomeric attachment between two microspheres held in optical traps (Fig. 1b, c, Extended Data Fig. 1b–d, Supplementary Video 1, Methods). This permitted us to handle and image chromosomes in a precisely controlled environment and accurately measure forces applied to the chromosome[24,25]. We visualized the trapped chromosomes using either wide-field fluorescence imaging of eGFP-labelled histone H2B, allowing the individual sister chromatids and the centromeres to be discerned (Fig. 1d), or

super-resolution imaging using binding-activated localization microscopy (BALM)[26], which permitted the visualization of individual fluorescent intercalators intermittently binding to a chromosome (Fig. 1e, Extended Data Fig. 1e). Our system is also compatible with multi-colour immunofluorescence analysis; for example, staining for histone H3 showed the expected pan-chromosome localization (Extended Data Fig. 1f), and staining for CREST and NCAPH revealed centromeres and chromosome scaffolds, respectively (Fig. 1f, Extended Data Fig. 1g, h, Methods). These different imaging strategies allowed us to confirm known features of chromosome organization.

## Mechanical properties of chromosomes

Optical manipulation enables mechanical features of chromosomes to be characterized with very high resolution. To achieve this, we first recorded force-extension curves by separating the two optical traps at a constant velocity of less than 0.2 μm s$^{-1}$ (Fig. 2a, b, Supplementary Videos 2, 3). For forces up to typically 10–50 pN, the force-extension behaviour of individual chromosomes was approximately linear, albeit with a large variability in stiffness between different chromosomes. By contrast, at higher forces, the chromosomes exhibited pronounced nonlinear stiffening, such that the force increased markedly with increasing chromosome extension. We determined chromosome length at the onset of this stiffening (stiffening length), which showed a broad distribution that is likely to reflect the known variability in the size of human chromosomes (2.5 ± 1.0 μm and 2.8 ± 1.7 μm for U2OS and HCT116 chromosomes, respectively, mean ± s.e.m.; Extended Data Fig. 2a). Moreover, the force-extension response was reversible at forces up to 300 pN (Fig. 2c, Extended Data Fig. 2b), as reported in micropipette aspiration studies[18,19].

To quantify the stiffening of chromosomes at high force, we determined the differential stiffness $K$ by evaluating the numerical derivative of the force $F$ with respect to the extension $d$ (Extended Data Fig. 2c, Methods). Beyond a critical force $F_c$ the differential stiffness increased following a power-law dependency; $K \sim F^\gamma$ (Fig. 2d), in which the stiffening exponent $\gamma$ characterizes how sensitive the stiffening is to force. Classical models for polymers predict a power-law stiffening, with the freely jointed chain (FJC) being characterized by the stiffening exponent $\gamma = 2$ and the worm-like-chain (WLC) by $\gamma = 3/2$ (refs. [4,27–29]). However, we observed a markedly weaker stiffening exponent compared to these classical models (Fig. 2d), with $\gamma = 0.82 ± 0.05$ for U2OS chromosomes. Despite a variability in initial stiffness between different chromosomes, all stiffness-force curves could be approximately collapsed onto a single master curve by scaling them to initial stiffness and critical force $F_c$ (Fig. 2e, Methods), which showed that the mechanical behaviour amongst chromosomes is consistent. To investigate whether this anomalous behaviour reflects structural remodelling under load—for example, owing to dynamic cross-linking such as that described for F-actin networks[30–32]—we measured the frequency dependence of the stiffening response. Using microrheology measurements, we quantified this viscoelastic response by applying a fixed pre-tension $F$ to chromosomes, and then a small-amplitude distance oscillation with frequency $\omega$ (refs. [28,32]). The differential force response is captured by the storage modulus $K'(F,\omega)$ and the loss modulus $K''(F,\omega)$, which characterize the elastically stored and the viscously dissipated mechanical energy, respectively (Methods). We determined that the storage modulus was constant over five orders of magnitude in frequency and was 10- to 100-fold larger than the loss modulus (Fig. 2f). When measurements were performed at varying pre-tensions, the storage modulus was consistent with the differential stiffness derived from the force-extension data (Fig. 2g). These results show that chromosomes have a predominantly elastic mechanical response over a wide range of frequencies, which is inconsistent with substantial dynamic remodelling occurring on these time scales. Thus, we attribute the notably weak power-law stiffening response to an intrinsic, nonlinear elastic response of the chromosome.

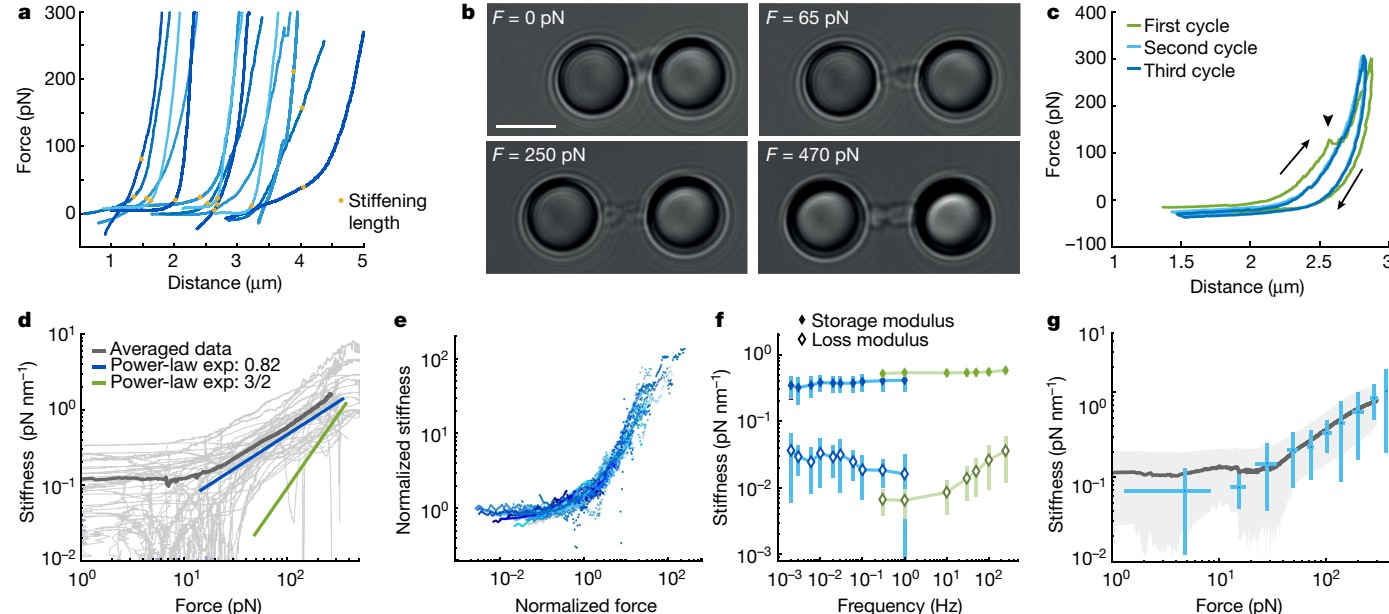

**Fig. 2 | Mechanical characterization of chromosome stretching.**
**a**, Representative series of force-extension curves depicting a linear stiffness regime for low forces ($F < 10$ pN) and a nonlinear stiffness regime at higher forces ($F > 20$ pN), in which the stiffness of the chromosomes increases with force. Stiffening lengths are indicated with dots. **b**, Representative bright-field images of stretching a U2OS chromosome. Scale bar, 4 μm. **c**, Three consecutive cycles of elongation and retraction (direction depicted by arrows). Abrupt declines in the force response (arrowhead) are suggestive of small rupture events and are most abundant during the first stretch cycle (23% of curves; $n = 155$ compared to 14% of third stretches; $n = 79$). **d**, Individual differential stiffness curves (light grey; $n = 44$) and average curve (dark grey) show a linear stiffness regime up to around 10 pN followed by power-law scaling in the regime between 20 and 200 pN, with a scaling exponent (exp) of $\gamma = 0.82 \pm 0.05$ (blue). The green line depicts scaling of 3/2 as expected for a WLC. **e**, Datasets show a universal form of the nonlinear stiffening after rescaling by $K_0$ and $F_c$, independent of the differences among chromosomes. **f**, The storage modulus (filled diamonds) and loss modulus (open diamonds) determined at a force of 50 pN with two methods of distance detection for the lower (blue; $n = 8$) and the higher (green; $n = 11$) frequencies (Methods). **g**, Oscillation data measured at 1 Hz ($n = 14$), overlapped with the differential stiffness derived from force-extension experiments (dark grey line, mean; light-grey shaded region, s.d.). In **f**, **g**, data represent mean values ± s.d. The data in **a**, **d**, **e**, **g** are from the third extension cycle.

## Chromosome hierarchical mechanics

Models of homogeneous polymers with finite extensibility $d_c$, such as the FJC or WLC, exhibit a nonlinear response at high force, such that $F \sim (d - d_c)^{-\delta}$, with $\delta > 0$ (refs. [4,27]). This divergent force-extension behaviour implies $K \sim F^\gamma$ with a stiffening exponent $\gamma = (\delta + 1)/\delta$, which is strictly larger than 1. Thus, the weak anomalous stiffening exponent ($\gamma < 1$) that we determined suggests that the nonlinear mechanical response lies in the inherent heterogenous nature of chromosomes. Heterogeneity of chromosomes is known to arise owing to the presence of specialized proteinaceous structures, such as centromeres, as well as the inherent differences between euchromatic and heterochromatic regions, which is reflected in the appearance of G-banding patterns after Giemsa staining[33,34]. To capture this heterogeneous nature of chromosomes, we propose a hierarchical worm-like chain (HWLC) model: an assembly with different structural elements represented by a series of WLCs with distinct contour and persistence lengths, which leads to an emergent nonlinear behaviour that is different from the response of the individual elements. Upon mechanical loading, these elements respond in a force-dependent hierarchy that leads to sequential stiffening (Fig. 3a–c, Extended Data Fig. 3). At low force, the response is dominated by the softest element. At higher forces, the stiffness of the next softest element dominates, and so on. The hierarchical nature of this model is characterized by the distribution $P(f_c)$ of internal critical forces of the individual elements. We compared our data to two classes of HWLC models: in one, we randomly drew the critical forces of the components from a power-law distribution $P(f_c) \propto f_c^{-\beta}$ (Fig. 3d); and, in the other, we drew them from an exponential distribution $P(f_c) \propto e^{-f_c/f_c^*}$ (Fig. 3e, Supplementary Note 1, Extended Data Fig. 4). Although both models showed anomalous stiffening, the power-law distribution offered a closer agreement to the spread in chromosome stiffness (Fig. 3d, e, Extended Data Fig. 5). The distribution of critical forces of the assembly for both models is in accord with experimental observations (Fig. 3f, Methods). However, only a power-law distribution for critical forces (Fig. 3d) provides a genuine power-law stiffening of the HWLC with an exponent, $\gamma = \beta + \alpha - 1$, in which $\alpha$ is set by the relation between the initial stiffness and the critical force of each element, $k_0 \propto f_c^\alpha$ (Supplementary Note 1). We conclude that the HWLC model can quantitatively account for the observed nonlinear mechanical response of human chromosomes.

## Mechanical role of TOP2A

To investigate the relationship between chromosome structure and mechanics, we depleted TOP2A, which is required for mitotic chromosome formation[3,11,35,36], and might also have a specific structural role[5,10,11,37]. We analysed chromosomes from human HCT116 CDK1as cells, in which TOP2A could be depleted during prometaphase using a 'degron' system[5] (Extended Data Fig. 6, Methods). We observed two populations of TOP2A-depleted chromosomes—one with a stiffening length equivalent to that of TOP2A-containing chromosomes, and another elongated and hypo-condensed population (Fig. 4a, b, Extended Data Fig. 7a). Hypo-condensed chromosomes were present at a higher frequency in chromosome spreads than in the tweezers (Extended Data Fig. 6f, Supplementary Note 2). Notably, depletion of TOP2A only led to minor quantitative changes in stiffening behaviour, which still followed the

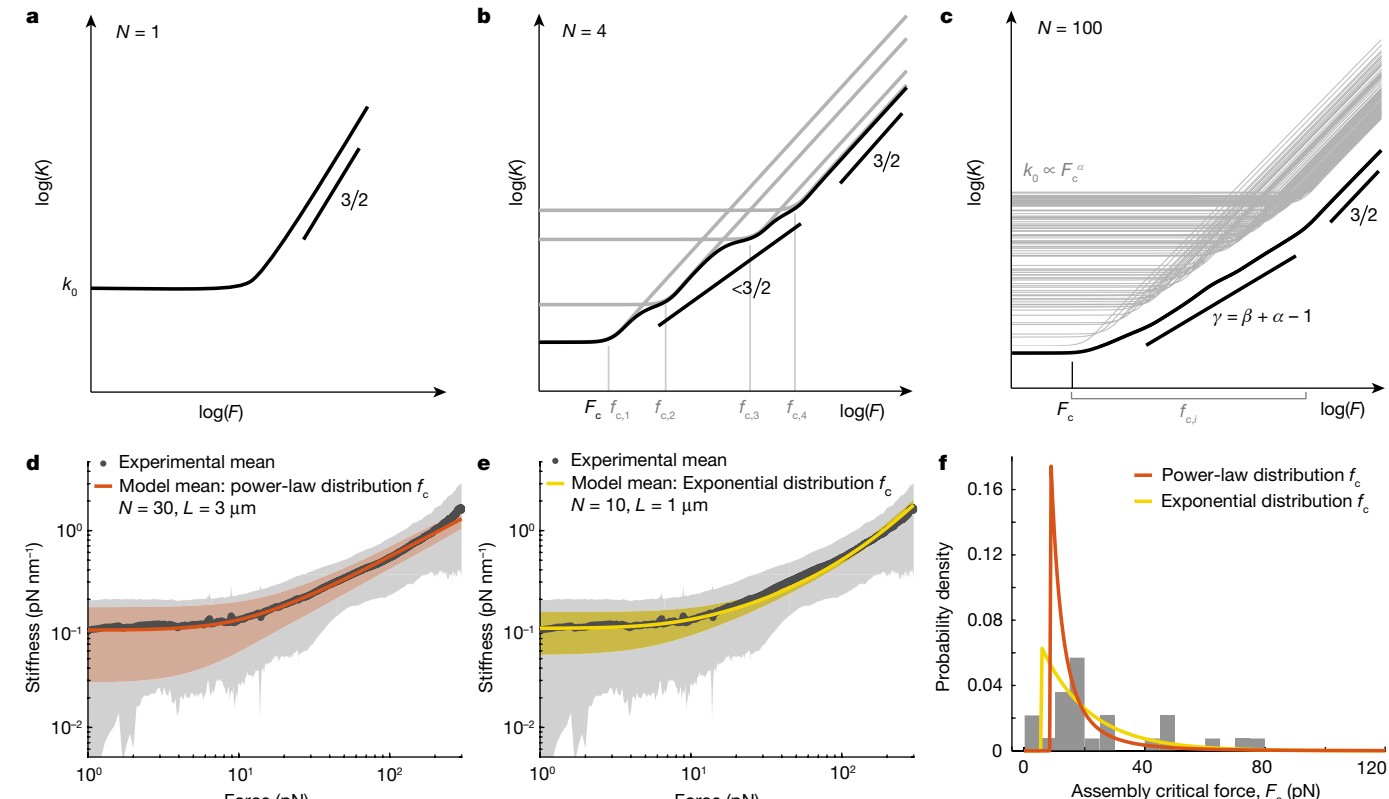

**Fig. 3 | Sequential stiffening of the HWLC model. a**, Stiffness-force curves of a single WLC transition from a constant stiffness directly towards a strain stiffening regime with a power-law scaling of 3/2. **b**, Stiffness-force curves of an assembly of four WLCs (individual WLC response in grey) in series result in an overall response (black) with an irregular transition zone to nonlinear stiffening that has a weaker stiffening behaviour than an individual WLC. The critical forces of the elements $f_{c,i}$ and of the assembly $F_c$ are indicated on the $x$ axis. **c**, Stiffness-force curve of an assembly (black) of 100 serial WLCs (individual WLC response in grey) with critical forces drawn from a power-law distribution,

$P(f_c) \propto f_c^{-\beta}$, resulting in a smooth power-law transition regime with an exponent of $\gamma = \beta + \alpha - 1$. **d**, **e**, Comparison of the experimental mean and s.d. of U2OS chromosomes (grey) with the mean and s.d. of the HWLC model, with $P(f_c) \propto f_c^{-\beta}$ with $\beta = 0.86$ ($N = 30$ and $L = 3 \mu m$) (orange) (**d**) and $P(f_c) \propto \exp(-f_c/f_c^*)$ with $f_c^* = 160$ pN ($N = 10$ and $L = 1 \mu m$) (yellow) (**e**). **f**, The theoretical distribution of assembly critical forces $F_c$ for the model parameters from **d** (orange) and **e** (yellow), superimposed with experimentally determined assembly critical forces of U2OS chromosomes ($n = 29$) (Methods).

HWLC model (Extended Data Fig. 7b, c, Methods). Fluorescence images of hypo-condensed, H2B–eGFP-labelled, TOP2A-depleted chromosomes showed a heterogeneous distribution of histones along the chromosome arms (Fig. 4b). After stretching, the brighter, chromatin-dense regions extended less than the less dense regions (Fig. 4c, Extended Data Fig. 7d), a heterogeneity consistent with our HWLC model.

To investigate whether TOP2A has a role in preserving the structure of condensed chromosomes, we induced chromosome decompaction (swelling) and re-compaction by alternating the KCl concentration[2,38] (Fig. 4d, Supplementary Video 4). During decompaction, we observed a strong elongation and softening of the chromosomes (Fig. 4e). After re-compaction, both control and TOP2A-depleted chromosomes returned to their initial length (Fig. 4f). Nevertheless, there was a significant increase in the compliance (inverse stiffness) of the TOP2A-depleted chromosomes after re-compaction (from $6 \pm 1$ nm pN$^{-1}$ to $15 \pm 3$ nm pN$^{-1}$ at 200 pN; $P = 0.0002$), unlike in control chromosomes (Fig. 4g, Methods). Similarly, we observed no change in the stiffening exponent of control chromosomes, whereas that of the hypo-condensed chromosomes (stiffening length greater than $5 \mu m$) decreased significantly from $1.1 \pm 0.2$ to $0.4 \pm 0.1$ ($P = 0.0017$) (Extended Data Fig. 7e). Within the HWLC framework, such a decrease in the stiffening exponent is interpreted as a structural perturbation with a flatter distribution $P(f_c)$ of the critical forces of the components. Moreover, the force-extension curves of chromosomes after decompaction did not change after prolonged exposure to high salt, indicating that there was no loss of material on the timescale of the

experiment (Extended Data Fig. 7f). We conclude that TOP2A assists in the restoration of the chromosome to its original structure after perturbation. Thus, TOP2A is not only indispensable for chromosome condensation, but it is also important for the preservation of mitotic chromosome structure.

## Conclusions

Here, we have introduced a strategy to study mitotic chromosomes using optical tweezers. We have successfully visualized individual human chromosomes at high resolution and analysed their mechanical parameters with a very high control of applied force. The chromosomes exhibited a nonlinear stiffening response with a power-law exponent considerably lower than that predicted by established polymer models. Although the initial stiffness and length of chromosomes was variable, the anomalous nonlinear stiffening was robust, suggesting that this is an inherent characteristic of chromosomes. This stiffening behaviour is distinct from the linear force-extension that has been reported previously using micropipettes[18,19], which probably stems from differences in force resolution and chromosome attachment. To explain the observed nonlinear mechanics, we developed a HWLC model based on sequential stiffening of hierarchical elements within a heterogeneous chromosome (Extended Data Fig. 3). Hence, the anomalous stiffening behaviour of a chromosome emerges from its intrinsic heterogeneity. It is tempting to speculate that such a hierarchy of nonlinear mechanical elements could be beneficial for maintaining

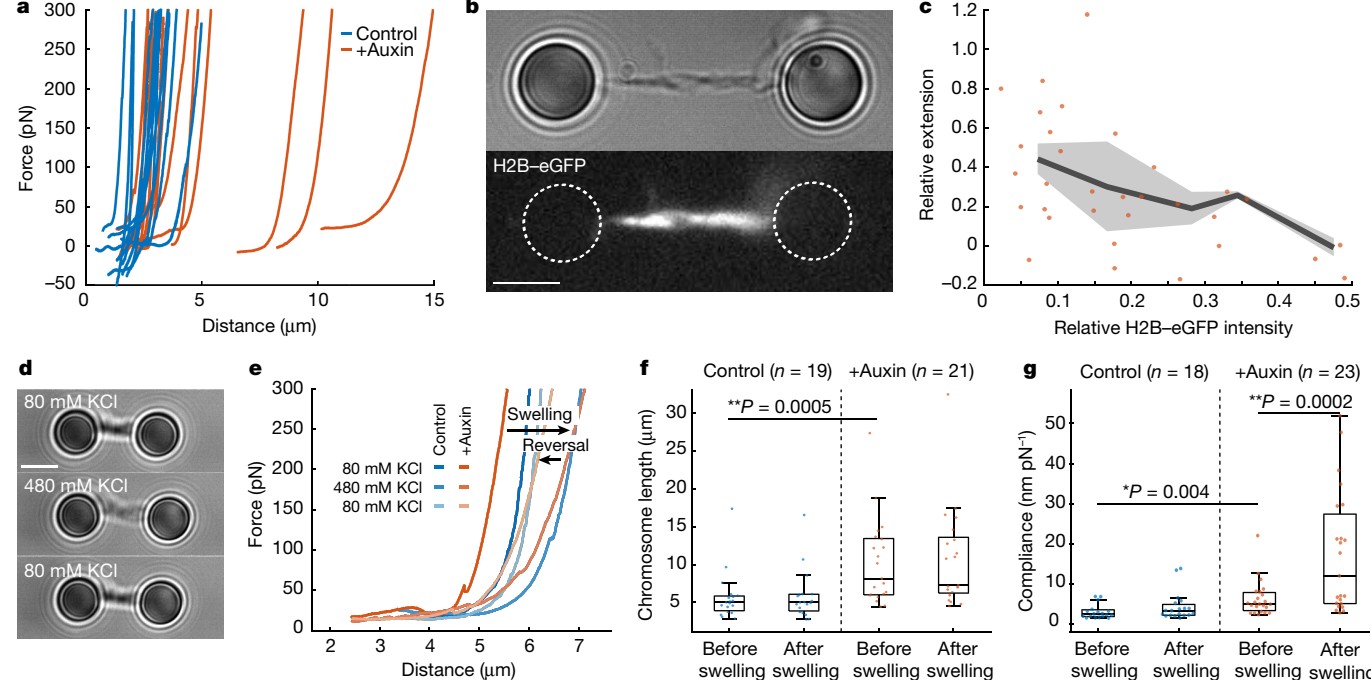

**Fig. 4 | Mechanical properties of TOP2A-depleted chromosomes.**
**a**, Stretching curves of control chromosomes (blue) and TOP2A-depleted chromosomes (orange). **b**, Representative bright-field image of a TOP2A-depleted chromosome and corresponding H2B immunofluorescence image. Scale bar, 4 μm. **c**, The relative extension of darker and brighter H2B–eGFP regions as a function of relative intensity (grey line and shaded region: mean ± s.e.m.). **d**, Representative bright-field images of a chromosome in its original buffer containing 80 mM KCl and subsequent images after flushing in buffer containing 480 mM KCl followed by buffer containing 80 mM KCl. **e**, Force-extension curves of a control and a TOP2A-depleted HCT116 chromosome before, during and after exposure to high-salt buffer (480 mM KCl). Changes in length (**f**) and compliance (**g**) of chromosomes before and after the exposing control and TOP2A-depleted chromosomes to high-salt buffer. Two-sided Wilcoxon rank-sum test, *P < 0.05; **P < 0.01. Centre, median; box, 25th to 75th percentile; whiskers, minimum and maximum data points (not considered as outliers).

the structural integrity of chromosomes by limiting the deformation of individual elements.

A future challenge is to identify the molecular basis for this HWLC model, and establish whether it could relate to patterns of heterochromatin and euchromatin[39], A-T and G-C content or the distribution of structural proteins along the chromosome[33,40,41]. A notable feature after depletion of TOP2A—the most abundant non-histone protein in metaphase chromosomes—is a reproducible shift in the stiffening response after perturbing chromosome structure. This is in stark contrast to other examples of elastic stress-stiffening polymer assemblies, in which non-destructive structural perturbations do not alter the stiffening exponent. Of note, the HWLC model can capture such a change in the stiffening exponent in terms of a structural modification. It has been hypothesized that TOP2A has a structural role by stabilizing chromatin loops through simultaneously binding two DNA duplexes in a closed gate[5]. Indeed, we find that TOP2A is essential to sustain chromosome mechanics after salt-induced chromosome expansion. The ability to manipulate and image chromosomes under controlled conditions makes our method suitable for investigating the structural and mechanical roles of other chromosomal proteins. We foresee that, by selective depletion or exposure to inhibitors, more insight will be obtained into how proteins compact and maintain mitotic chromosomes. In summary, we have transformed an already-powerful single-molecule technique into a quantitative and versatile method for investigating the mitotic chromosome.

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

# Methods

## Cell lines and cell culture

All cell lines were cultured in DMEM supplemented with 10% fetal bovine serum (FBS) and penicillin–streptomycin in a humidified incubator at 37 °C and 5% $CO_2$. Unless indicated otherwise, all cell lines were obtained from and authenticated by the ATCC by karyotyping and STR profiling. The U2OS TRF1-BirA cell line[22] was a gift from R. J. O' Sullivan and was authenticated by karyotyping. Endogenous H2B in the U2OS TRF1-BirA cell line was tagged with eGFP, as described previously[5]. To rapidly deplete TOP2, we used a HCT116 TOP2A-mAID cell line that also expressed H2B–eGFP, facilitating chromosome identification. The HCT116 TOP2A-mAID H2B-eGFP cell line was a gift from D. F. Hudson and authentic by karyotyping, and was described in a previous report[5]. All cell lines were routinely tested for mycoplasma and shown to be negative. To achieve tight temporal control over cell synchrony, the HCT116 TOP2A-mAID H2B-eGFP cell line was modified for CDK1as chemical genetics by knock-in of CDK1as and knockout of endogenous CDK1, as described previously[42] and based on another previous report[43]. The constructs for CDK1as were gifts from W. Earnshaw (Addgene 118596 and 118597) and from Z. Izsvak (Addgene 34879). We determined that treatment with 0.25 µM 1NM-PP1 (529581, Sigma-Aldrich) for 16 h efficiently arrests HCT116 TOP2A-mAID H2B-eGFP CDK1as cells at the G2−M boundary (Extended Data Fig. 6a, b). The incubation time and concentration of 1NM-PP1 were optimized by propidium iodide flow cytometry (Extended Data Fig. 5a), performed as described before[5]. Efficient release from the arrest was achieved with two wash cycles by centrifugation with preheated medium. A Neon transfection system (Thermo Fisher Scientific) was used for transfections of HCT116 and U2OS cell lines according to the manufacturer's recommendations. 1NM-PP1 and nocodazole were purchased from Sigma-Aldrich. The synthetic auxin indole-3-acetic acid (IAA) sodium salt (sc-215171, Santa Cruz) was used. Six days before chromosome isolation, HCT116 TOP2A-mAID CDK1as cells were transduced with lentiviruses introducing TRF1-BirA into the genome. These cells were treated for 16 h with 0.25 µM 1NM-PP1, before release into 100 ng ml$^{-1}$ nocodazole (Sigma-Aldrich) with or without 500 µM auxin for 4 h, to arrest cells in prometaphase and deplete TOP2A, respectively. Mitotic cells were detached by shaking and chromosomes were isolated from this population (Extended Data Fig. 6c). Chromosome spreads were performed as described previously[5] and showed an altered chromosome morphology following exposure to auxin in accordance with what was reported[5]. Approximately 75% of TOP2A depleted chromosomes appeared hypocondensed compared to 5% of control chromosomes (Extended Data Fig. 6d). Immunostaining of TOP2A on chromosome spreads was not detectable in auxin-treated samples, confirming efficient depletion of TOP2A (Extended Data Fig. 5e).

## Lentiviral production and transduction

Third-generation lentiviral particles were generated for integration of BirA-TRF1. HEK293T cells were grown with 25 µM chloroquine diphosphate (Sigma-Aldrich) for 5 h before being transfected with plasmids pMD2.G, pMDLg/pRRE and pRSV-Rev (Addgene 12259, 12251 and 12253, deposited by D. Trono[44]) and a transfer plasmid for BirA-TRF1 integration. A Calphos mammalian transfection kit (Clontech) was used for transfections according to the manufacturer's protocol. Eighteen hours after transfection, the medium was replaced with fresh medium. Forty-eight hours after transfection, the growth medium was collected and centrifuged at 500g for 5 min, and the supernatant containing viral particles was filtered through a 0.45-µm membrane before being concentrated 10× using an Amicon Ultra-15 100 kDa centrifugal unit (Merck-Millipore). The viral concentrate was snap-frozen and stored at −80 °C. For lentiviral transduction, a T-75 flask of 75% confluent HCT116 TOP2A-mAID CDK1as cells was incubated with 7.5 µg ml$^{-1}$ polybrene in 3 ml 10× lentiviral concentrate

and 7 ml growth medium for 1 h with mixing every 15 min. Cells were then seeded in a T-175 flask and the culture was expanded before chromosome isolation.

## Chromosome isolation

A previously reported method, with modifications, was used to isolate mitotic chromosomes in large quantities with minimal contamination with cell debris[23]. In brief, cells were grown with 12.2 mg l$^{-1}$ biotin (Sigma-Aldrich) for 24 h before isolation. On the day of isolation, 8–10 T175 flasks of cells were treated for 4 h with 200 ng ml$^{-1}$ nocodazole (Sigma-Aldrich) and then mitotic shake-off was used to enrich for mitotic cells, resulting in $1 \times 10^7$–$2 \times 10^7$ mitotic cells. The mitotic cells were centrifuged at 300g for 5 min, resuspended in 10 ml 75 mM KCl and 5 mM Tris-HCl (pH 8.0) and then incubated for 10 min at room temperature. All subsequent steps were carried out at 4 °C. Cells were centrifuged at 300g for 5 min and then resuspended in 8 ml polyamine (PA) buffer (15 mM Tris-HCl (pH 8.0), 2 mM EDTA, 0.5 mM EGTA, 80 mM KCl, 20 mM NaCl, 0.5 mM spermidine, 0.2 mM spermine and 0.2% Tween-20) for U2OS cells and a PA* buffer (15mM Tris-HCl (pH 7.4), 0.5 mM EDTA-K, 80 mM KCl, 1 mM spermidine, 0.4 mM spermine and 0.1% Tween-20) for HCT116 cells, both supplemented with Complete mini protease and PhosSTOP phosphatase inhibitor cocktails (Roche). This suspension was then lysed in a Dounce homogenizer using 25 strokes with a tight pestle. The suspension was cleared twice of cell debris by centrifugation at 300g for 5 min. Chromosomes were purified using a glycerol step gradient containing two layers (60% and 30% glycerol in PA). After centrifugation at 1,750g for 30 min, the chromosomes were collected from the 60% glycerol fraction and stored at −20 °C in around 60% glycerol in PA buffer at a concentration of $10^6$–$10^7$ chromosomes per ml. Chromosomes could be stored for up to two months without undergoing any noticeable change in mechanical properties.

## Dual trap optical tweezers with wide-field fluorescence

The dual trap optical set-up was described previously[24]. In brief, two optical traps were created using a 20 W, 1064 nm CW fibre laser (YLR-20-LP-IPG, IPG Photonics). Two traps were created by splitting the laser beam into two paths using a polarizing beam splitter cube and could be steered independently using one accurate piezo mirror (Nano-MTA2X10, Mad City Labs) and one coarse positioning piezo step mirror (AG-M100N). After the two paths were recombined, they were coupled into a Nikon microscope body using two 300 mm lenses, and focused in the flow cell with a 1.2 NA water immersion objective (Nikon, Plan apo VC NA1.2). Back-focal plane interferometry was used to measure forces, and bead tracking was performed by LED illuminated bright-field imaging on a CMOS camera (DCC1545M, Thorlabs). Wide-field epifluorescence was achieved by illumination with 488, 532, 561 and 639 nm lasers (Cobolt 06-01 Series) and detection by separation of the emission light using an OptoSplit III (Cairn Research) and imaging on an EMCCD camera (iXon 897 Life, Andor Oxford Instruments Technology).

## Microfluidics and flow cell preparation

A microfluidic flow-system (u-Flux, LUMICKS B.V.) was used to insert solutions into a five-channel flow cell (LUMICKS B.V.; Fig. 1b). Before each experiment, bleach cleaning was performed to remove residual debris from flow cell, followed by sodium thiosulfate neutralization. Passivation was performed to reduce chromosome attachment to tubing and flow cell walls by incubation for 1 h with 0.05% casein solution, followed by excessive rinsing with PA buffer. Chromosomes diluted in PA buffer (10–20 µl in 500 µl) were inserted into a side channel of the flow cell (Fig. 1b). Streptavidin-coated polystyrene microspheres (diameter: 4.6 µm, Spherotech) in PA buffer (4 µl in 300 µl) were inserted in one of the main channels. Other channels were filled with PA buffer unless stated otherwise.

## Chromosome attachment and force-extension

To facilitate attachment of the biotinylated chromosome between two streptavidin-coated microspheres (diameter: 4.6 μm), one trapped microsphere was brought into the proximity of a chromosome in solution, resulting in attachment of the telomeric end of the chromosome to the microsphere (Fig. 1b, Extended Data Fig. 1b). Next, the microspheres were moved to another microfluidic channel and fluid flow was activated (Fig. 1b). The chromosome attached to one of the microspheres was flow-stretched to confirm correct attachment (Extended Data Fig. 1b, c) and then brought into the proximity of the other microsphere to induce attachment of the other chromosome end (Fig. 1c). Note that owing to the relatively small cross-section of chromosomes compared to the microspheres, both telomeric ends from one sister chromatid would attach to the microsphere occasionally. Non-biotinylated chromosomes showed only very limited attachment to the microspheres (Extended Data Fig. 1d).

## Immunofluorescence

Chromosomes were incubated overnight at 4 °C with primary antibody in a concentration of 5 μg ml$^{-1}$ and were subsequently diluted fivefold in PA buffer and stored for 1 h at 4 °C. Next, chromosomes were incubated with secondary antibody in a concentration of 5 μg ml$^{-1}$ for 1 h at room temperature. After addition of PA buffer to dilute the sample again by fivefold, chromosomes were stored for 30 min at 4 °C. To remove excess antibody, chromosomes were centrifuged at 750$g$ for 5 min on a 20 μl glycerol cushion. The supernatant was then removed, leaving around 100 μl chromosome solution that could be used for imaging. Primary antibodies were anti-NCAPH (1:100, HPA002647, Sigma Aldrich), CREST anti-sera (1:200 HCT-0100, Immunovision), anti-TRF2 (1:100, sc-9143, Santa Cruz), anti-H3S10 (1:400, 06-570, Sigma-Aldrich) and anti-H3-Alexa Fluor 647 (1:200, 15930862, Thermo Fisher Scientific). Secondary antibodies were anti-rabbit IgG-Alexa Fluor 647 (1:500, A-21244, Thermo Fisher Scientific), anti-rabbit IgG-Alexa Fluor 568 (1:500, A-11011, Thermo Fisher Scientific) and anti-human IgG-Alexa Fluor 488 (1:500, A-11013, Thermo Fisher Scientific). Biotinylated TRF1 was detected using streptavidin–Alexa Fluor 568 (1:200, S11226, Invitrogen).

## Immunoblotting

SDS–PAGE and immunoblotting was performed as described previously[5]. In brief, cell pellets were lysed in RIPA buffer containing cOmplete Mini EDTA free (Roche) on ice for 20 min. Samples were then sonicated in a water-cooled Bioruptor Pico (Diagenode) and centrifuged at 21,000$g$ for 15 min at 4 °C. Protein concentration was determined using a Pierce BCA protein assay kit (Thermo Fisher Scientific). Forty micrograms of protein was loaded per well. The primary antibodies were anti-CDK1 (1:1,000, ab133327, Abcam), anti-Myc (1:1,000, sc-40, Santa Cruz) and anti-histone H3.3 (1:5,000, ab176840, Abcam). The secondary antibodies were anti-mouse IgG peroxidase conjugate (1:10,000, A4416, Sigma-Aldrich) and anti-rabbit IgG peroxidase conjugate (1:10,000, A6154, Sigma-Aldrich).

## Determination of differential stiffness, stiffening length and compliance

To calculate the differential stiffness from force-distance curves, the force distance curve was first smoothed using a moving average with a window size of 1/15 of the total data points in the force curve, followed by numerical differentiation of force with respect to distance. To determine the onset of stiffening, the plateau stiffness was determined as the most likely stiffness at forces below 50 pN, as estimated from the maximum of a kernel density estimate of the stiffness values. The onset of stiffening was then determined as the point at which the stiffness exceeds the plateau stiffness by one standard deviation of all stiffnesses below 50 pN. To determine the compliance at 200 pN, the inverse of the stiffness of the chromosome at a force of 200 pN was used.

## Collapse of stiffness-force curves

To achieve a collapse of the stiffness-force curves they were normalized on a log-log-scale. Therefore, curves were interpolated to a logarithmic force scale to get evenly spaced data after taking the logarithm. In addition, negative values for force and stiffness were discarded. Then the logarithms of force ln($F$) and stiffness ln($K$) were calculated and fitted with a piecewise function $y = \ln(K_0)$ for $x \le \ln(F_c)$ and $y = c - \ln(F_c) + \ln(K_0)$ for $x > \ln(F_c)$ to determine the initial stiffness $K_0$ and the critical force $F_c$. If the determined parameters for $K_0$ and $F_c$ were in the range of the stiffness-force curve, the curves normalized by $K_0$ and $F_c$ were plotted in a double-logarithmic plot to achieve the collapse. The criteria that $K_0$ and $F_c$ had to be positive and within the range of the stiffness-force curve were met by 29 out of 44 curves.

## Microrheology

Oscillations of the optical trap were generated by applying a sinusoidal voltage to the analogue input of the piezo mirror controller (Nano-Drive, MCL) to apply the oscillation on top of the digitally controlled mirror position. The signal was first generated digitally using Labview (National Instruments). The analogue voltage was then produced with a digital analogue converter (DAQ, National Instruments). Oscillations were produced with an amplitude of 25 mV corresponding to a trap displacement of roughly 200 nm. The frequency of the oscillation was varied between 2 mHz and 100 Hz. When the frequency was varied in the experiment, the pre-tension was kept constant at 50 pN (Fig. 2f). Experiments for different pre-tension were performed with a frequency of 0.1 Hz (Fig. 2g). To avoid limitations by the frame rate of the bead tracking camera at higher frequencies (>1 Hz), the bead position at high frequencies was calculated from the force and the trap position (Fig. 2f, green line). Data analysis of the oscillations was performed in MATLAB (Mathworks). First, the bead–bead distance and the force were synchronized on the basis of the position of the stationary bead where the force was measured, based on the cross-correlation between the bead position from bead tracking and the measured force. Then the oscillatory data were analysed following a previously described procedure[45]. In brief, both the force and the bead–bead separation were detrended and fitted with a sine function with a fixed frequency set to the experimental frequency and an additional offset. Then the complex stiffness was calculated as $k = \frac{A_F}{A_d} e^{i(\varphi_F - \varphi_d)}$ with the amplitude and the phase of the force oscillation $A_F$ and $\varphi_F$, and the amplitude and the phase of the distance oscillation $A_d$ and $\varphi_d$, respectively.

## Calculating HWLC force responses

Model curves in Fig. 3d, e were constructed by first defining a distribution for each system parameter. For simplicity, the number of sub-chains, $N$, and the length of each sub-chain, $L/N$, were kept constant. A power-law or exponential distribution with cut-offs was chosen for the sub-chain critical force, $f_c$. Given these distributions, we analysed the responses of 500 HWLC configurations, each constructed by sampling $N$ values of $f_c$. The force-response of each configuration was computed by summing the extensions of each sub-chain at a given force, found by numerically solving the flexible WLC equation (Supplementary Note 1). The force-response curve was then numerically differentiated, and the mean and standard deviation of the stiffness-force curve were compared to experimental data.

## Distribution of $F_c$

The distribution of the critical force of a HWLC assembly, $F_c$, corresponds to the force at which its softest element starts stiffening. Hence, for a given distribution, $P(f_c)$, $F_c$ is distributed as the minimum of $N$ independent samples. Its cumulative distribution function (CDF) satisfies $P(F_c \le x) = 1 - (1 - P(f_c \le x))^N$. This expression can be differentiated to yield the probability density function (PDF) ,$P(F_c = x) = N P(f_c = x) \quad (1 - P(f_c \le x))^{N-1}$. Figure 3f shows $P(F_c = x)$ for

the power-law distribution with cut-offs and for an exponential distribution with cut-offs. The experimental critical forces were determined as described above.

## Statistics and reproducibility

Average values and errors were represented as mean ± s.e.m. unless indicated otherwise. Differences in populations are tested using a two-sided Wilcoxon rank-sum test, where $P < 0.05$ is regarded as significant (*) and $P < 0.01$ as highly significant (**). The sample sizes for representative microscopy images are as follows: Fig. 1c $n = 91$, Fig. 1d–f $n = 3$, Fig. 2b $n = 91$, Fig. 4b $n = 5$, Fig. 4d $n = 20$, Extended Data Fig. 1a $n = 24$, Extended Data Fig. 1b, c $n = 3$, Extended Data Fig. 1e $n = 10$, Extended Data Fig. 1f–h $n = 3$, Extended Data Fig. 6a $n = 2$, Extended Data Fig. 6e (amount of cells) $n = 117$ (15 min), $n = 120$ (20 min), $n = 98$ (25 min), $n = 121$ (30 min), $n = 110$ (40 min), $n = 115$ (50 min), Extended Data Fig. 6g $n = 16$ (control), $n = 19$ (+auxin), Extended Data Fig. 7d $n = 5$.

## Reporting summary

Further information on research design is available in the Nature Research Reporting Summary linked to this paper.

## Data availability

The data supporting the findings in this study are openly available from the Dataverse repository at https://doi.org/10.34894/XFZZPJ.

## Code availability

The Julia code to calculate the force response of the HWLC is openly available from the GitHub repository accessible at https://doi.org/10.5281/zenodo.5970943.

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

**Acknowledgements** We thank I. Samejima and W. Earnshaw for providing training in chromosome isolation; R. O'Sullivan for the U2OS TRF1-BirA cell line; S. Acar for help with cell culture; V. Bjerregaard for help with the lentiviral transduction; and A. Biebricher for help with optical tweezers. This work was supported by European Union Horizon 2020 grants (Chromavision 665233 to G.J.L.W., I.D.H., E.J.G.P. and Y.L.; and Antihelix 859853 to Y.L. and I.D.H.), the European Research Council under the European Union's Horizon 2020 research and innovation program (MONOCHROME, grant agreement no. 883240 to G.J.L.W.), the Novo Nordisk Foundation (NNF18OC0034948 to I.D.H. and G.J.L.W.), the Deutsche Forschungsgemeinschaft (WI 5434/1-1 to H.W.), the Dutch Research Council (NWO Vidi 640-47-555 to I.H.), the Nordea Foundation (to I.D.H.) and the Danish National Research Foundation (DNRF115 to Y.L. and I.D.H.).

**Author contributions** G.J.L.W., I.D.H., E.J.G.P., Y.L. and I.H. conceived the study. A.E.C.M., E.K., G.H.H. and H.W. performed the optical tweezers experiments. K.S., C.F.N. and A.H.B. developed cell lines and performed biochemical analysis of chromosomes. A.E.C.M., H.W., C.P.B. and G.J.L.W. performed data analysis and interpretation. J.H. and C.P.B developed the HWLC model. A.E.C.M., H.W., C.F.N., J.H., C.P.B., E.J.G.P., I.D.H. and G.J.L.W. wrote the manuscript. G.J.L.W., I.D.H., E.J.G.P and Y.L. led the research and the interpretation of the results. All authors discussed the results and commented on the manuscript.

**Competing interests** G.J.L.W., E.J.G.P. and I.H. own shares of LUMICKS. G.J.L.W. and E.J.G.P. serve on the technical advisory board of LUMICKS.

**Additional information**
**Correspondence and requests for materials** should be addressed to Chase P. Broedersz, Ian D. Hickson or Gijs J. L. Wuite.

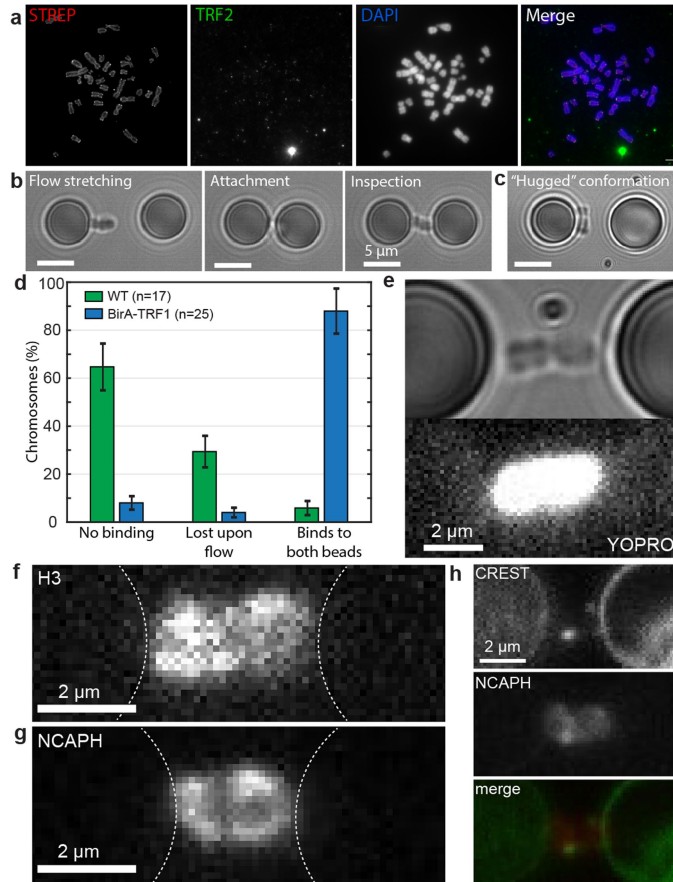

**Extended Data Fig. 1 | Biotin labelling, attachment and imaging of mitotic chromosomes. a**, Representative images of chromosome spreads from U2OS cells stained with Alexa Fluor 568 Streptavidin, telomere repeat-binding factor 2 (TRF2) and DAPI to show an overlap between the Streptavidin foci and TRF2 staining at the telomeres. **b**, Representative bright-field images of chromosome attachment to two microspheres using flow-stretching. **c**, Approximately 20% of chromosomes became attached to the first microsphere in a 'hugged' conformation (along the length of a single sister chromatid), probably as a result of the binding of the telomeres from opposite ends of the chromatid to one microsphere. In these cases, the chromosome and bead would be rejected either immediately or following the inevitable failure to attach a second bead during attempted chromosome flow-stretching. **d**, Quantification of attachment efficiency of chromosomes to microspheres from wild-type U2OS cells and BirA-TRF1 expressing U2OS cells (mean values ± s.e.m.) **e**, Representative images showing fluorescence of the intercalator YOPRO and a corresponding bright-field image. Small circular dark spots in panels **c** and **e** are a result of dust on optical surfaces outside of the flow cell. **f**, **g**, Representative images of immuno-staining of H3 histones (**f**) and NCAPH (**g**). **h**, Immuno-staining images for NCAPH and CREST to show localization of condensin I along the chromatids and one focus that reveals the position of a centromere. The second centromere is not located in the focal plane and therefore is not visible.

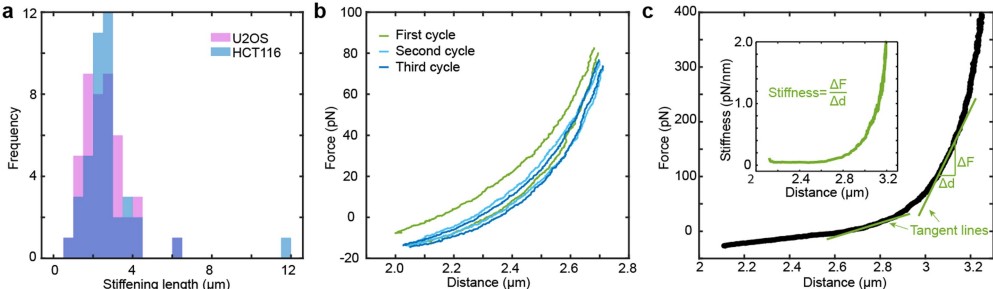

**Extended Data Fig. 2 | Chromosome length, repeated stretching and stiffness calculation. a**, Histogram of stiffening length of U2OS and HCT116 chromosomes. **b**, Three consecutive elongation and retraction curves of a chromosome stretched to around 80 pN. **c**, Differential stiffness was determined by taking the numerical derivative of the force-extension data; that is, approximating the slope of a tangent line to the data. Stiffness-distance plots (inset) showing that the stiffness increases with distance.

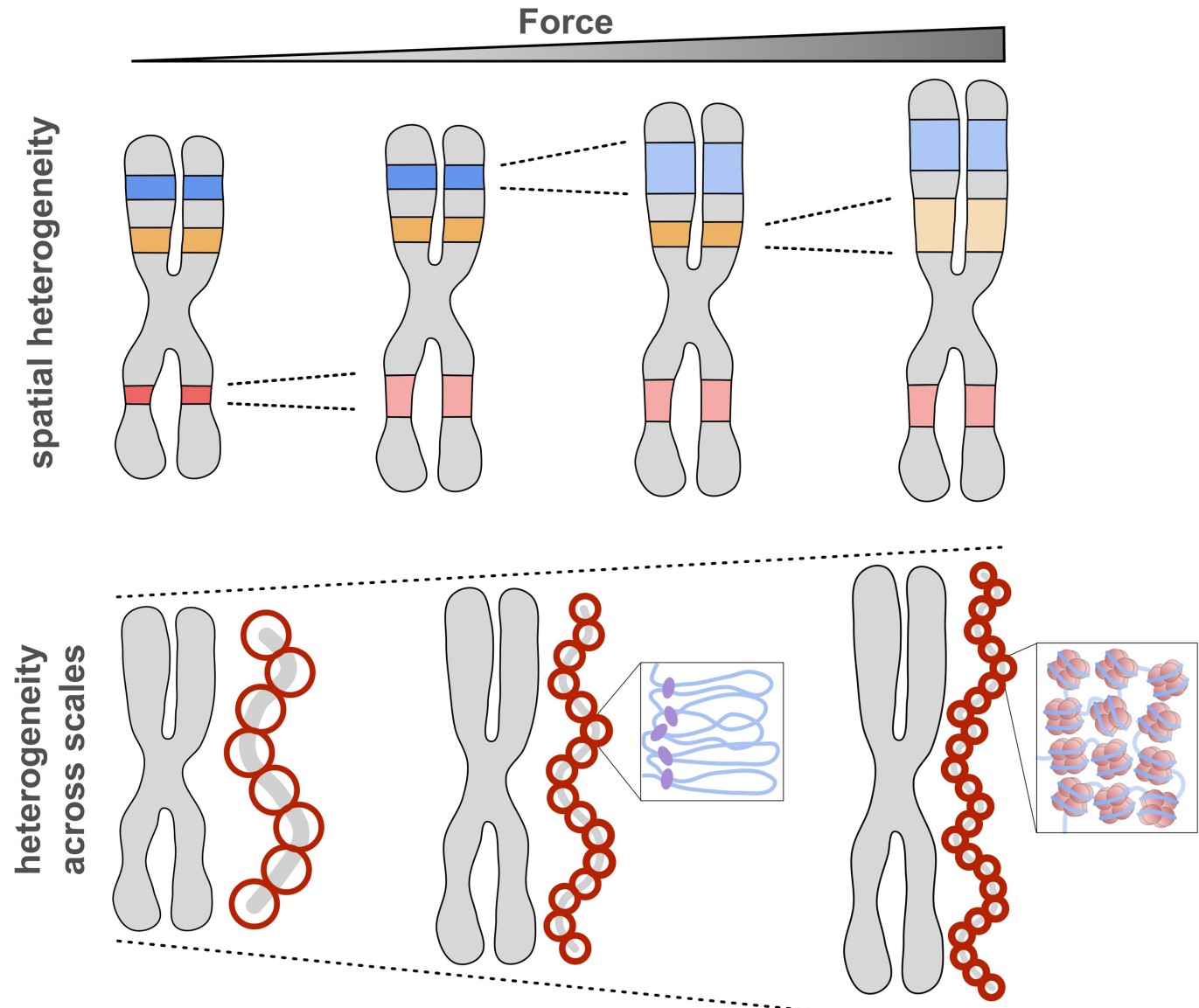

**Extended Data Fig. 3 | Scheme of the HWLC.** Schematic depicting how the Hierarchical Worm-Like Chain (HWLC) captures the mechanical response of a chromosome as a function of force. The HWLC captures two distinct contributions of chromosome heterogeneity: 1) spatial heterogeneity (top row), where different chromosomal regions (depicted in red, orange and blue) exhibit distinct mechanical properties, and 2) scale-dependent heterogeneity (bottom row), where chromosomal structure has distinct mechanical properties on different length-scales (schematically depicted as red circles showing WLC elements of different scales, increasing stiffness at smaller scales is indicated by thicker lines) relating to different structural elements: examples shown here speculatively are chromatin loops and nucleosomes. These levels of chromosome heterogeneity are represented in the HWLC as a serial assembly of WLC elements with distinct linear stiffnesses and critical stiffening forces. The nonlinear mechanics of the HWLC emerges from the sequential stiffening of these underlying elements, reflecting how different chromosomal regions and length-scales dominate the mechanical response of the chromosome as the force is varied.

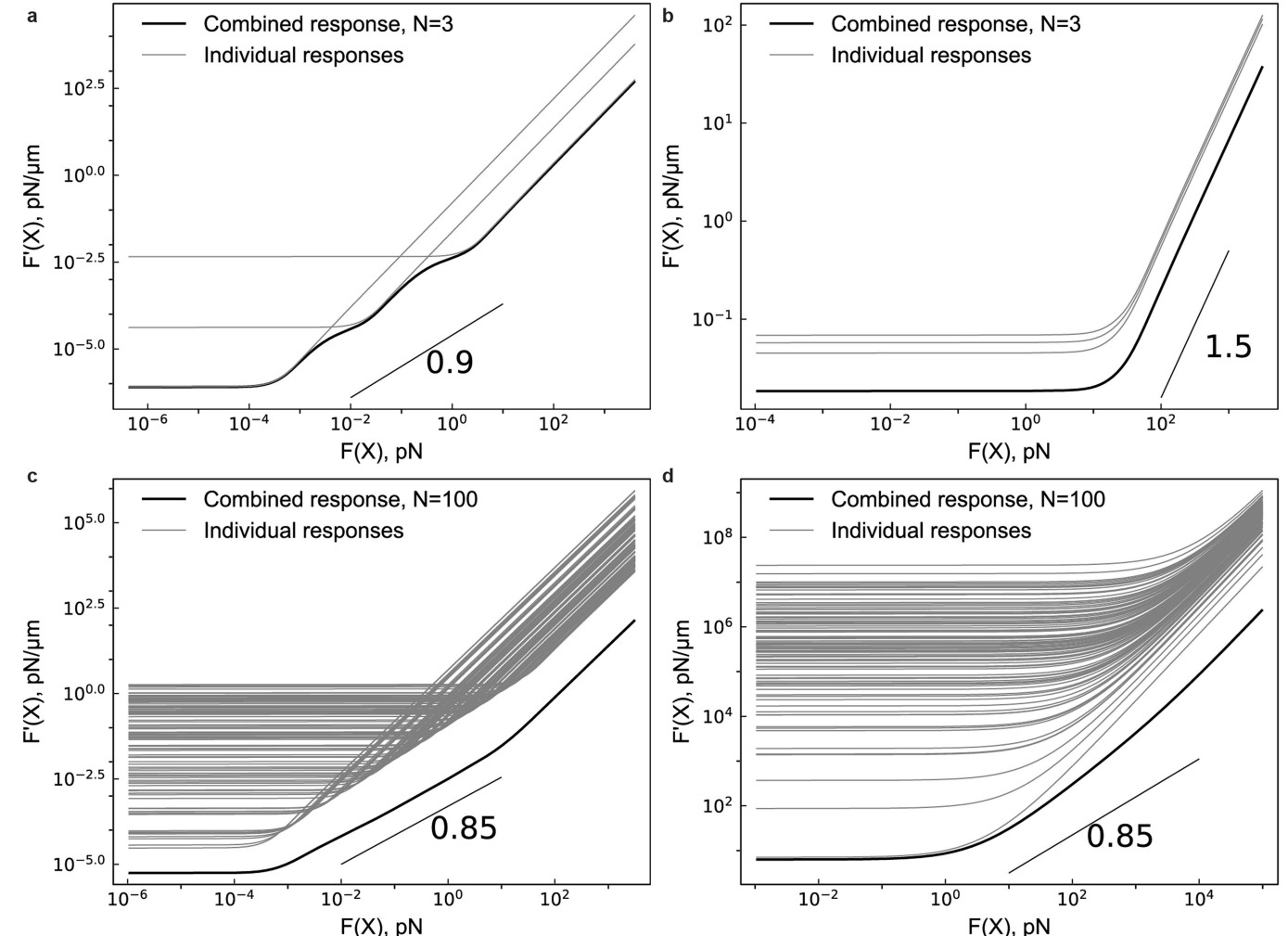

**Extended Data Fig. 4 | Individual and total response of WLC assemblies.**
**a**, The individual and total responses for an assembly of three flexible WLCs. For this example, critical forces are given by a power-law; $P(f_c) \propto f_c^{-0.9}$, with cut-offs 0.0004 and 40 pN. All chains are of equal length $L = 1,000$ nm, and the constitutive law is $k_0 = L^{-1} f_c$. **b**, The individual and total response curves for an assembly of three flexible WLCs, with power-law distributed critical forces $P(f_c) \propto f_c^{-1}$, with cut-offs 0.0004 and 40 pN. The length of each component is chosen as $l(f_c) = (3,000/(\sum f_{c,i}^{-2}))f_c^{-2}$, so that the total length of the assembly is

3,000 nm, and the constitutive law is $k_0(f_c) = l(f_c)^{-1} f_c \propto f_c^3$. **c**, The individual and total response curves for an assembly of 100 flexible WLCs with power-law distributed critical forces; $P(f_c) \propto f_c^{-\beta}$, $\beta = 0.85$, and equal component lengths; $k_0(f_c) = l^{-1} f_c^\alpha$, $\alpha = 1$, $l = 30$ nm. Slope prediction is given by $\beta + \alpha - 1 = 0.85$. Cut-offs for $P(f_c)$ are 0.0004 and 40 pN. **d**, The individual and total response curves for an assembly of 100 semi-flexible WLCs with an exponential distribution of critical forces: $P(f_c) = (e^{-f_c/50})/50$. The linear spring coefficients are $k_0 = f_c^2 \times 90/(\pi^4 k_B T)$.

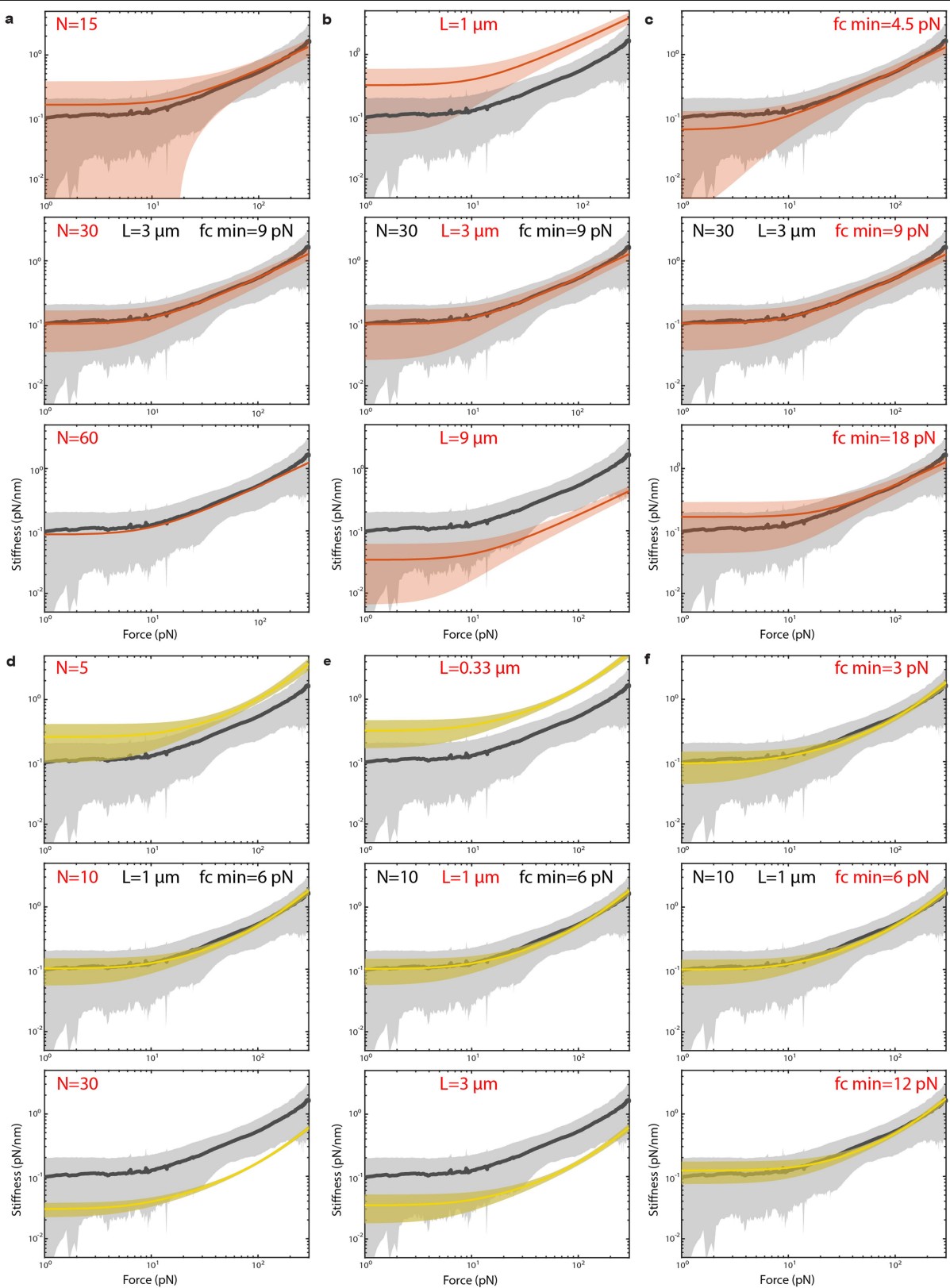

**Extended Data Fig. 5 | Effect of model parameters on the HWLC.** The panels in each vertical column show the effect of varying one model parameter (the number of sub-chains, $N$, the chromosome length, $L$, or the lower cut-off for the critical force, $f_{c,min}$) on the distribution of stiffness curves for the power-law (**a**–**c**) and exponential distribution (**d**–**f**). Other parameters are kept at the same values as in the middle row. **a**, **d**, The number of sub-chains affects the initial stiffness of the assembly and sets the scale of the variance of stiffnesses.

**b**, **e**, The stiffness of the assembly varies inversely with its total length. **c**, **f**, The lower cut-off for the distribution of critical forces sets the initial stiffness of the ensemble and the force at which it starts to stiffen. Line and shaded regions depict mean and standard deviation of experimental data of U2OS chromosomes (grey), HWLC model with $P(f_c) \propto f_c^{-\beta}$ (orange) and HWLC model with $P(f_c) \propto \exp(-f_c/f_c^*)$ (yellow).

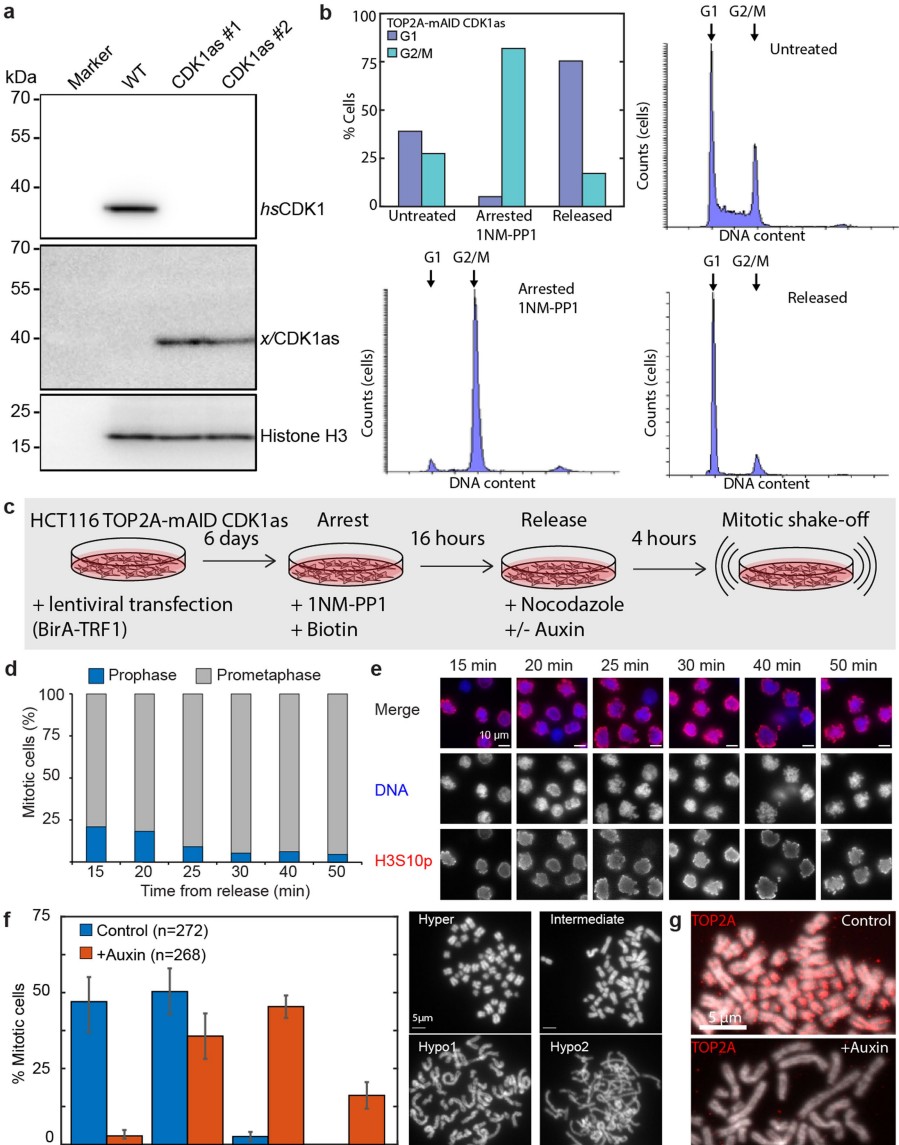

**Extended Data Fig. 6 | Characterization of cell synchronization and TOP2A depletion. a**, Immunoblot of wild-type HCT116 cells (WT) and two clones of CDK1as-modified cells, in which the genes encoding the 35 kDa endogenous hsCDK1 has been knocked out (top panel) and the mutant CDK1as gene has been knocked-in and is expressed ectopically (*xl*CDK1as; 40 kDa; middle panel). *xl*CDK1as is not recognized by the hsCDK1 antibody and was detected using an antibody to the c-myc epitope tag. #1 is a TOP2A-mAID clone and #2 the TOP2A-mAID H2B-EGFP clone used in this study. Histone H3 was used as a loading control (bottom panel). For gel source data, see Supplementary Fig. 1. **b**, Optimization of CDK1as cell synchronization. Quantification of G1 and G2/M content from Propidium Iodide-flow cytometry profiles of untreated HCT116 TOP2A-mAID CDK1as cells, HCT116 TOP2A-mAID CDK1as cells grown with 0.25 μM 1NM-PP1 for 16 h to arrest them in G2 phase, and HCT116 TOP2A-mAID CDK1as cells grown with 0.25 μM 1NM-PP1 for 16 h and then released from this

block and grown for 4 h. **c**, Diagram of the experimental procedure for TOP2A depletion. **d**, Quantification of mitotic cells at the indicated time points following release from CDK1as arrest, as depicted in (**c**). **e**, Representative immunofluorescent images of cells released from CDK1as arrest at the indicated timepoints shown in (**d**). pH3S10 antibody (red) was used as a mitotic marker and DAPI (blue) was used to stain DNA. **f**, Quantification of chromosome morphology into four categories (based on degree of condensation) from chromosome spreads in control versus TOP2A degraded cells with representative images of the categories on the right (error bars: s.e.m.). Values are the mean of three independent experiments with a total count as mentioned in the figure. **g**, Representative images of immunostaining for TOP2A on chromosome spreads of control and the auxin-exposed HCT116 TOP2A-mAID CDK1as cells to confirm degradation of TOP2A in the auxin-exposed chromosomes.

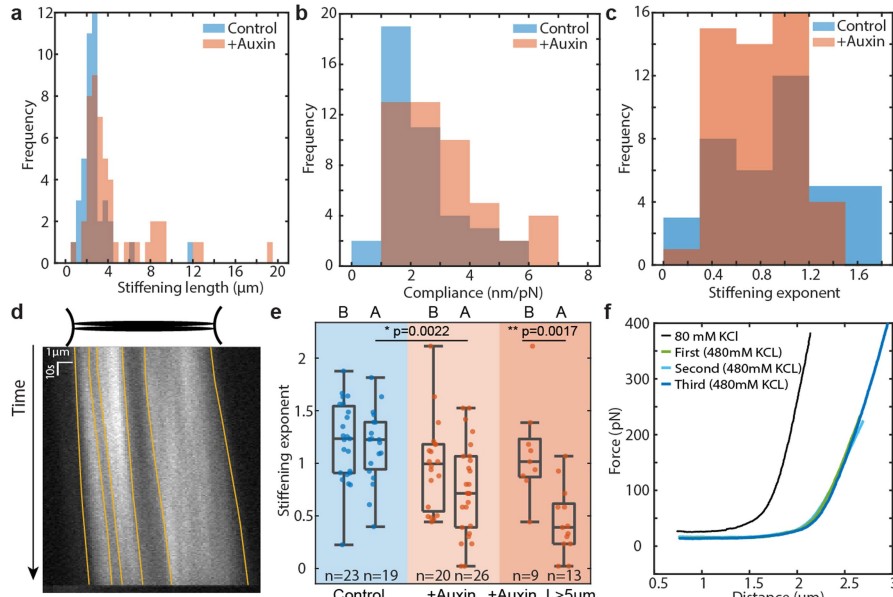

**Extended Data Fig. 7 | Additional characterization of the mechanics of TOP2A-depleted chromosomes. a**–**c**, Histograms of stiffening length (**a**), compliance at 200 pN (**b**) and strain stiffening exponent (**c**) for control and auxin-exposed (TOP2A-depleted) chromosomes. **d**, Kymograph of fluorescence signal during stretching of hypo-condensed, H2B–eGFP-labelled, TOP2A-depleted chromosomes. Above the kymograph, the initial position of the chromosome between microspheres is schematically illustrated. The boundaries of regions with different histone intensity signal are depicted by lines. **e**, Stiffening exponent of force-extension curves for HCT116 control chromosomes, auxin-exposed chromosomes and auxin-exposed chromosomes with a stiffening length above 5 µm before (B) and after (A) salt-induced decompaction. Two-sided Wilcoxon rank-sum test, * $p < 0.05$; ** $p < 0.01$. Centre: median, box: 25th to 75th percentile, whiskers: minimum and maximum datapoints (not considered as outlier). **f**, Force-extension curve of an HCT116 chromosome before salt-induced decompaction (maintained at 80 mM KCl; black line) and three consecutive force-extension curves during salt induced decompaction at 480 mM KCl.

# Reporting Summary

## Statistics

For all statistical analyses, confirm that the following items are present in the figure legend, table legend, main text, or Methods section.

| n/a | Confirmed | |
|---|---|---|
| ☐ | ☒ | The exact sample size (*n*) for each experimental group/condition, given as a discrete number and unit of measurement |
| ☐ | ☒ | A statement on whether measurements were taken from distinct samples or whether the same sample was measured repeatedly |
| ☐ | ☒ | The statistical test(s) used AND whether they are one- or two-sided<br>*Only common tests should be described solely by name; describe more complex techniques in the Methods section.* |
| ☒ | ☐ | A description of all covariates tested |
| ☒ | ☐ | A description of any assumptions or corrections, such as tests of normality and adjustment for multiple comparisons |
| ☐ | ☒ | A full description of the statistical parameters including central tendency (e.g. means) or other basic estimates (e.g. regression coefficient) AND variation (e.g. standard deviation) or associated estimates of uncertainty (e.g. confidence intervals) |
| ☐ | ☒ | For null hypothesis testing, the test statistic (e.g. *F*, *t*, *r*) with confidence intervals, effect sizes, degrees of freedom and *P* value noted<br>*Give P values as exact values whenever suitable.* |
| ☒ | ☐ | For Bayesian analysis, information on the choice of priors and Markov chain Monte Carlo settings |
| ☒ | ☐ | For hierarchical and complex designs, identification of the appropriate level for tests and full reporting of outcomes |
| ☒ | ☐ | Estimates of effect sizes (e.g. Cohen's *d*, Pearson's *r*), indicating how they were calculated |

*Our web collection on statistics for biologists contains articles on many of the points above.*

## Software and code

Policy information about availability of computer code

| Data collection | Custom made LabVIEW (2011 SP1) software was used to obtain optical tweezers data |
|---|---|
| Data analysis | Custom made MATLAB (2020a) scripts were used to analyse the data, simulations of HWLC were preformed using custom written Julia (1.6) scripts |

For manuscripts utilizing custom algorithms or software that are central to the research but not yet described in published literature, software must be made available to editors and reviewers. We strongly encourage code deposition in a community repository (e.g. GitHub). See the Nature Portfolio guidelines for submitting code & software for further information.

## Data

Policy information about availability of data

All manuscripts must include a data availability statement. This statement should provide the following information, where applicable:
- Accession codes, unique identifiers, or web links for publicly available datasets
- A description of any restrictions on data availability
- For clinical datasets or third party data, please ensure that the statement adheres to our policy

The data supporting the findings in this study are openly available from the DataverseNL repository at https://doi.org/10.34894/XFZZPJ.

# Field-specific reporting

Please select the one below that is the best fit for your research. If you are not sure, read the appropriate sections before making your selection.

☒ Life sciences ☐ Behavioural & social sciences ☐ Ecological, evolutionary & environmental sciences

For a reference copy of the document with all sections, see nature.com/documents/nr-reporting-summary-flat.pdf

# Life sciences study design

All studies must disclose on these points even when the disclosure is negative.

| | |
|---|---|
| Sample size | No sample size calculations were performed. The sample sizes are indicated in the method section/figure captions and follow typical values used in similar micromechanical studies. |
| Data exclusions | Data was excluded if chromosomes detached during the experiment or if other experimental errors occurred such as air bubbles moving inside the flow chamber or experimental drift due to drying of water on the objective. |
| Replication | The experiments were performed spread over the course of more than one year, measured on more than 5 different biologically independent samples and performed by different experimenters showing consistent results. All replications were successful except for the exclusion criteria detailed above. |
| Randomization | This does not apply in our case since the experimental protocol inherently guarantees random selection of chromosomes. |
| Blinding | No blinding was applied during data collection or analysis. However, standardized procedures for data collection and analysis were used to prevent bias. |

# Reporting for specific materials, systems and methods

We require information from authors about some types of materials, experimental systems and methods used in many studies. Here, indicate whether each material, system or method listed is relevant to your study. If you are not sure if a list item applies to your research, read the appropriate section before selecting a response.

## Materials & experimental systems

| n/a | Involved in the study |
|---|---|
| ☐ | ☒ Antibodies |
| ☐ | ☒ Eukaryotic cell lines |
| ☒ | ☐ Palaeontology and archaeology |
| ☒ | ☐ Animals and other organisms |
| ☒ | ☐ Human research participants |
| ☒ | ☐ Clinical data |
| ☒ | ☐ Dual use research of concern |

## Methods

| n/a | Involved in the study |
|---|---|
| ☒ | ☐ ChIP-seq |
| ☒ | ☐ Flow cytometry |
| ☒ | ☐ MRI-based neuroimaging |

## Antibodies

| | |
|---|---|
| Antibodies used | Primary antibodies: Anti-NCAPH (1:100, HPA002647, Sigma Aldrich), CREST anti-sera (1:200 HCT-0100, Immuno-vision), anti-TRF2 (1:100, sc-9143, Santa Cruz), anti-H3S10 (1:400, 06-570, Sigma-Aldrich) and anti-H3-Alexa Fluor 647 (1:200, 15930862, ThermoFisher), anti-CDK1 (1:1000, ab133327, Abcam), anti-Myc (1:1000, sc-40, Santa Cruz) anti-histone H3.3 (1:5000, ab176840, Abcam).<br>Secondary antibodies: anti-rabbit IgG-Alexa Fluor 647 (1:200, A-21244, ThermoFisher), anti-rabbit IgG-Alexa Fluor 568 (1:200, A-11011, ThermoFisher) and anti-human IgG-Alexa Fluor 488 (1:200, A-11013, ThermoFisher), anti-mouse IgG peroxidase conjugate (1:10000, A4416, Sigma-Aldrich) and anti-rabbit IgG peroxidase conjugate (1:10000, A6154, Sigma-Aldrich). |
| Validation | HPA002647 (NCAPH) was validated by IHC and WB, and was used in the Human Protein Atlas and in Zhan et al. 2018 Oncol Rep and Scott et al. 2017 Mol Syst Biol. HCT-0100 antisera (CREST) was validated by ELISA and used in Nielsen et al. 2020 PNAS. sc-9143 (TRF2) was used in Liu at al. 2017 Cell and 26 other publications, but validation data is not available from the company. 06-570 (H3S10) was validated by ICC, WB and IP and used in Otsuki and Brand 2018 Science and more than 100 other publications. 15930862 (H3-AF 647) was validated by ICC/IF. ab133327 (CDK1) was validated by WB, IP and IHC-P, and used in Valeri-Alberni et al. 2021 Cell Rep and 43 other publications. Sc-40 (MYC) was validated by WB, IHC and IF, and published in Arany et al. 1993 Viral Immunol and Seo et al. 2021 Nat Comm. ab176840 (H3.3) was validated by ChIP, WB, IF and IHC and used in Udugama et al. 2021 Nat Comm and 16 other publications.<br>For details see manufacturer websites. |

# Eukaryotic cell lines

| | |
|---|---|
| Cell line source(s) | The parental U2OS cells were obtained from and authenticated by ATCC (USA). They were modified into U2OS BirA-TRF1 cells and authenticated like described in Garcia-Exposito et al. 2016, Cell reports. The parental HCT116 CMV-TIR1 cells [Natsume et al. 2016, Mol Cell] to the HCT116 TOP2A-mAID cells and obtained from and authenticated by RIKEN, BRC Cell bank in Tsukuba, Japan. The HCT116 CMV-TIR1 cells were modified into HCT116 TOP2A-mAID and HCT116 TOP2A-mAID H2B-EGFP cells and authenticated by PCR, Immunoblotting and Sanger sequencing like described in Nielsen et al. 2020, PNAS. HCT116 TOP2A-mAID and HCT116 TOP2A-mAID H2B-EGFP cells were routinely grown with triple antibiotic selection (blastidicin, hygromycin, puromycin) to retain purity. HEK293T cells were obtained from and authenticated by ATCC (USA). |
| Authentication | The ATCC authenticated the wild type U2OS, HCT116 and HEK293T cell lines used in this study by STR profiling. |
| Mycoplasma contamination | All cell lines were tested negative for mycoplasma contamination |
| Commonly misidentified lines (See ICLAC register) | No commonly misidentified cell line was used in the study. |

