## [Peer Review File · Nature]

Manuscript Title: Nonlinear mechanics of human mitotic chromosomes

Reviewer Comments & Author Rebuttals

Reviewer Reports on the Initial Version:

Referee #1:

This manuscript describes a combination of force spectroscopy, wide-field and immunofluorescence imaging on individual chromosomes. Previous work manipulated chromosomes using micropipettes and found that condensin is key to the stiffness of these structures. Here, a clever labeling technique fuses biotin to telomeres located on the chromosome ends, allowing high-resolution optical tweezers to be used on such a construct for the first time. A single fixed chromosome may be seen in bright field images (including the centromere), in super-resolution fluorescence images, and in immune-staining to reveal the localization of condensin within the chromosome. As seen before, the elastic response of the chromosome is complex and highly variable. A new model, the Hierarchical Worm-like Chain (HWLC), is introduced, to explain non-linear stiffening seen under tension due to the heterogeneous nature of chromosome structure. Finally, it is shown that TOP2A-depleted chromosomes are not only less stiff, but less able to reform after the release of tension.

This concise and well-written paper introduces and carefully explains several intriguing techniques that could provide answers to many outstanding questions about the chromosome. The results here are of great interest to the readers of the journal. After addressing a few minor questions, it should be suitable for publication.

Specific Questions:

Fig. 1: Why is there such a difference in the bright field images between panels c) and d)? Is this due to the fluorophore binding to the beads or some other effect?

Fig. 1 and Ext. Data Fig. 1: How difficult is it to trap a chromosome and place it at the imaging focal plane? What are the experimental consequences of an out-of-plane image and attachment? Could this explain some of the variability in the force-extension curves?

Fig. 1 and Methods: When both ends of a sister chromatid bound to the same bead, how was this detected? Did differences appear in the force-extension curves?

Fig. 3f: The data fitting that discriminates between the two models fits to a histogram with effectively only three bins. Could these fits be also tested with a narrower bin width?

Fig. 4f,g: The stars shown here presumably indicate confidence intervals. But could this be stated specifically?

Results, p. 5 and Discussion: The conclusion that “TOP2A provides a structural memory for refolding of the chromosome to its original structure after perturbation” is perhaps too sweeping, as this isn’t really demonstrated. The data presented only convincingly shows that some structure with a similar stiffness is restored after perturbation, and not necessarily the original structure. This is still an interesting result that yields new information about the role of TOP2A, so it is not necessary to overstate the findings.

Referee #2:

This work addresses the internal structure of mitotic chromosomes and outstanding questions over the role of TOP2A. In metaphase chromosomes, when compaction and shaping have largely taken place, the protein remains concentrated at the axis of each sister chromatid. This has prompted questions over whether this protein’s enzyme activity has a role to play in maintaining, or introducing further catenation, within metaphase chromosomes. Alternatively, does it have other roles, as part of the chromosome scaffold, possibly in acting as a protein clamp or linker (as has been suggested for condensin) or could it act to recruit other factors to mitotic chromosomes?

The contribution of this protein is hard to dissect: the interpretation of phenotypes seen in vertebrate cells genetically modified to allow for extensive, but slow depletion, is complicated by the accumulation of effects arising from perturbations throughout the cell cycle. A rapid decrease in activity can be achieved through the use of inhibitors/poisons, but these produce confounding side effects, such as DSBs, or artificial drug-protein clamps. In this study, use has been made of a HCT116-derived cell line generated previously by Christian Nielsen and Damien Hudson, in which the endogenous TOP2A alleles have been tagged with an auxin-inducible degron, allowing for rapid removal (within an hour) of most of the protein. Nielsen and colleagues (2020) used this cell line to show, through fixed and live imaging, that loss of TOP2A from chromosomes already in prometaphase/metaphase affects compaction.

In the present study, new sophisticated chromosome manipulation and imaging techniques have been applied to shed further light on the internal structure of mitotic chromosomes and on the role of TOP2A in their maintenance. Metaphase chromosomes have been isolated using conditions that aim to preserve the native structure, and prepared for examination using optical tweezers, super-resolution fluorescence microscopy (BALM) and microfluidics. These methods used reveal novel nonlinear stiffening properties. An intriguing new model is proposed, wherein chromosome structures behave like hierarchical worm-like chains. This is a departure from the linear force-extension found by John Marko and colleagues using micropipette aspiration and is not consistent with the helical staircase conformation previously proposed. Additional experiments provide further evidence that TOP2A plays a role in maintenance of chromosome compaction during mitosis. The figures are well presented and appropriate statistical tests have been used. This is exciting work that addresses an important area for research and would likely be of general interest. However, some areas need to be addressed.

Comments/ suggested improvements:

1. This report contains new methods that will almost certainly advance our understanding of mitotic chromosomes; however, it remains a largely descriptive study. The new findings concerning physical properties of chromosomes (stiffness) are not related to biological structures or molecular biology. Instead, the authors discuss (235) “softer elements” and (240) “the weakest element”. While the figures are well presented, with clear microspheres and mitotic chromosomes, Fig. 5 is not informative - the coils do not relate to the chromosome arms. The authors do not attempt to discover the molecular basis of the chains, and there is no indication of scale. Perhaps a visual correlation of changes in size of chromosomal banding could be correlated with phase of stretching? Indeed, the authors comment (242) that: “A future challenge will be to identify molecular...elements in this...model (using) G-banding patterns or the distribution of structural proteins along the chromosome.”

(191-197) Auxin-inducible depletion of TOP2A resulted in cells with hypo-condensed chromosomes. There were only minor changes in stiffening behaviour. These hypo-condensed chromosomes had altered H2B-eGFP-labelling. Interestingly, the brighter areas extended less than the dimmer regions. Perhaps this gives a clue to the scale of the chains? This should be reviewed in the Discussion.

If possible, incorporation of G banding data (or other underlying structural information) into the study would be of broad interest and would strengthen the MS.

2. In the manuscript it is stated that “through inducible degradation of TOP2A in cells with already condensed mitotic chromosomes, we provide support for a role of TOP2A in the maintenance of a compacted chromosome structure.” However, the protocol as outlined in Ext. Data Fig. 4 would result in TOP2A levels decreasing as cells enter M phase, rather than being degraded from chromosomes already in prometa/metaphase. Given that cells released from the arrest will not move into M phase completely synchronously, in some cells chromosomes may already have undergone extensive compaction before TOP2A levels fall, while others will be compacting chromosomes where the protein has been largely removed. Therefore, the population examined will be very heterogeneous, which may explain the two populations of TOP2A-depleted chromosomes the authors describe: a hypocondensed population, which may reflect cells where compaction has occurred in the absence of TOP2A (given that lack of TOP2A has been shown by others to result in a failure of condensin to compact chromosomes beyond a thin ribbon-like state, however long they remain arrested in nocodazole), while those with a more normal appearance may be from cells in which chromosome compaction, through the action of both condensin I and II, had taken place prior to TOP2A degradation.

The effect of TOP2A removal on the maintenance of metaphase chromosomes would be better discerned by releasing cells from G2/M arrest into nocodazole 1 hr before the addition of auxin, with chromosomes then sampled at ~2 hr +/- auxin. In addition, to control for the fact that, in cells expressing TOP2A, chromosomes will continue to compact while arrested in nocodazole, the structure of chromosomes from which TOP2A has been degraded should be compared with TOP2A-containing chromosomes sampled at the point when auxin was added into the culture (rather than with controls sampled after further arrest in nocodazole).

3. In this reviewer's view, it is uncertain whether the data from the swelling and re-compactation expts (Fig. 4) can be used to argue for TOP2A providing a structural memory. Since TOP2A is a major non-histone protein component of the mitotic chromosome, the obvious criticism is that its removal may simply give rise to perturbation of structure arising from extensive depletion of protein making up the chromosome, rather than reflecting a specific role of TOP2A (for example, a role as an internal linker, as suggested for condensin). One way to address this might be to examine the impact of catalytically inactive forms of TOP2A. While expressing an enzymatically-dead form of the enzyme in living cells may not be feasible (because of dominant-negative effects), this could be looked at through incubation of isolated chromosomes with purified protein prior to structural analysis. If the effects on structure, detected when chromosomes have been prepared from auxin-treated cells, can be restored by incubating isolated chromosomes with purified human WT TOP2A (-/+ ATP or with a nonhydrolyzable ATP analogue), this rescue could be compared to that seen when using other versions of TOP2, such as TOP2B, or mutant forms of TOP2A (e.g. catalytically-dead Y805S, or catalytically-perturbed forms such as K662A, or a version deleted for the C-terminal Chromatin Tether Domain, that has been shown to be required for TOP2A to associate robustly with mitotic chromatin). These expts would help to determine whether it is the presence of catalytically-active TOP2A that is required, and whether other activities of this protein contribute to metaphase structure.

Minor comment:

Some of the terms used within the MS are not widely used in biology labs, and a brief explanation within the text might save readers time checking definitions. These include: storage modulus, loss modulus, stiffening exponent.

Author Rebuttals to Initial Comments:

Referee #1:

This manuscript describes a combination of force spectroscopy, wide-field and immunofluorescence imaging on individual chromosomes. Previous work manipulated chromosomes using micropipettes and found that condensin is key to the stiffness of these structures. Here, a clever labeling technique fuses biotin to telomeres located on the chromosome ends, allowing high-resolution optical tweezers to be used on such a construct for the first time. A single fixed chromosome may be seen in bright field images (including the centromere), in super-resolution fluorescence images, and in immunostaining to reveal the localization of condensin within the chromosome. As seen before, the elastic response of the chromosome is complex and highly variable. A new model, the Hierarchical Worm-like Chain (HWLC), is introduced, to explain non-linear stiffening seen under tension due to the heterogeneous nature of chromosome structure. Finally, it is shown that TOP2A-depleted chromosomes are not only less stiff, but less able to reform after the release of tension.

This concise and well-written paper introduces and carefully explains several intriguing techniques that could provide answers to many outstanding questions about the chromosome. The results here are of great interest to the readers of the journal. After addressing a few minor questions, it should be suitable for publication.

We thank the referee for their very positive evaluation of our work.

Specific Questions:

Fig. 1: Why is there such a difference in the bright field images between panels c) and d)? Is this due to the fluorophore binding to the beads or some other effect?

The bright-field images in figure 1c show the chromosome completely in focus, while in figure 1d it is slightly out-of-focus. In our optical tweezers' setup, the focal plane for fluorescence imaging can be adjusted independently of the focus of the trapping laser and brightfield imaging. In figure 1d the trapping height was adjusted such that the chromosome was in the focal plane for the fluorescence imaging, which caused it to be out of focus for bright field imaging, leading to a different appearance. We added a short statement for clarification in the figure legend.

Fig. 1 and Ext. Data Fig. 1: How difficult is it to trap a chromosome and place it at the imaging focal plane? What are the experimental consequences of an out-of-plane image and attachment? Could this explain some of the variability in the force-extension curves?

Capturing a chromosome is quite straightforward. The initial attachment of the bead is facilitated by the trapping laser. Since the trapped beads are not rotationally constrained, the attached chromosome will automatically move into the imaging plane during flow-stretching (3rd image in figure 1b) and remain in this plane once it is attached to the second bead. Hence, this is not effect that leads to variability in the force-extension curves. We have modified the legend to figure 1 accordingly.

Fig. 1 and Methods: When both ends of a sister chromatid bound to the same bead, how was this detected? Did differences appear in the force-extension curves?

Correct attachment of a bead is readily discernible by visual inspection, as can be appreciated from the image in extended data figure 1c. On the other hand, if a microsphere becomes attached to both ends of the same sister chromatid and this is detected by visual inspection, then the chromosome and bead would be discarded. However, on the rare occasions when we might overlook such an attachment, the chromosome would not be able to bind to the second bead and again would be rejected. We have added a short statement for clarification in the legend to extended data figure 1.

Fig. 3f: The data fitting that discriminates between the two models fits to a histogram with effectively only three bins. Could these fits be also tested with a narrower bin width?

Following the referee's suggestion, we explored different binning widths. While the comparison is showing that a different binning does not change the interpretation of the data, it became apparent that the intermediate binning is arguably a more insightful way to present the data. We updated figure 3f accordingly.

We would like to add that figure 3f is not showing a fit of the (binned) experimental data, but a prediction of the distribution of critical forces based on the parameters optimized to describe the stiffnesses presented in figure 3d/e. Therefore, the binning does not influence the determination of the model and the data in figure 3f are consistent with both models. However, the experimental data in figures 3d/e are better reproduced with a power-law distribution, as we note in the main text. We rephrased the manuscript to make this more clear.

Fig. 4f,g: The stars shown here presumably indicate confidence intervals. But could this be stated specifically?

We have now added the information in the figure legend, which was previously only found in the method section.

Results, p. 5 and Discussion: The conclusion that “TOP2A provides a structural memory for refolding of the chromosome to its original structure after perturbation” is perhaps too sweeping, as this isn't really demonstrated. The data presented only convincingly shows that some structure with a similar stiffness is restored after perturbation, and not necessarily the original structure. This is still an interesting result that yields new information about the role of TOP2A, so it is not necessary to overstate the findings.

We agree with the referee that the term “structural memory” is too sweeping to be supported by our data. We therefore rephrased this section of the text and toned down our conclusions.

Referee #2:

This work addresses the internal structure of mitotic chromosomes and outstanding questions over the role of TOP2A. In metaphase chromosomes, when compaction and shaping have largely taken place, the protein remains concentrated at the axis of each sister chromatid. This has prompted questions over whether this protein's enzyme activity has a role to play in maintaining, or introducing further catenation, within metaphase chromosomes. Alternatively, does it have other roles, as part of the chromosome scaffold, possibly in acting as a protein clamp or linker (as has been suggested for condensin) or could it act to recruit other factors to mitotic chromosomes?

The contribution of this protein is hard to dissect: the interpretation of phenotypes seen in vertebrate cells genetically modified to allow for extensive, but slow depletion, is complicated by the accumulation of effects arising from perturbations throughout the cell cycle. A rapid decrease in activity can be achieved through the use of inhibitors/poisons, but these produce confounding side effects, such as DSBs, or artificial drug-protein clamps. In this study, use has been made of a HCT116-derived cell line generated previously by Christian Nielsen and Damien Hudson, in which the endogenous TOP2A alleles have been tagged with an auxin-inducible degron, allowing for rapid removal (within an hour) of most of the protein. Nielsen and colleagues (2020) used this cell line to show, through fixed and live imaging, that loss of TOP2A from chromosomes already in prometaphase/metaphase affects compaction.

In the present study, new sophisticated chromosome manipulation and imaging techniques have been applied to shed further light on the internal structure of mitotic chromosomes and on the role of TOP2A in their maintenance. Metaphase chromosomes have been isolated using conditions that aim to preserve the native structure, and prepared for examination using optical tweezers, super-resolution fluorescence microscopy (BALM) and microfluidics. These methods used reveal novel nonlinear stiffening properties. An intriguing new model is proposed, wherein chromosome structures behave like hierarchical worm-like chains. This is a departure from the linear force-extension found by John Marko and colleagues using micropipette aspiration and is not consistent with the helical staircase conformation previously proposed. Additional experiments provide further evidence that TOP2A plays a role in maintenance of chromosome compaction during mitosis. The figures are well presented and appropriate statistical tests have been used. This is exciting work that addresses an important area for research and would likely be of general interest. However, some areas need to be addressed.

We would like to thank the referee for their very positive evaluation of our work

Comments/ suggested improvements:

1. This report contains new methods that will almost certainly advance our understanding of mitotic chromosomes; however, it remains a largely descriptive study. The new findings concerning physical properties of chromosomes (stiffness) are not related to biological structures or molecular biology. Instead, the authors discuss (235) "softer elements" and (240) "the weakest element". While the figures are well presented, with clear microspheres and mitotic chromosomes, Fig. 5 is not informative - the coils do not relate to the chromosome arms. The authors do not attempt to discover the

molecular basis of the chains, and there is no indication of scale. Perhaps a visual correlation of changes in size of chromosomal banding could be correlated with phase of stretching? Indeed, the authors comment (242) that: “A future challenge will be to identify molecular...elements in this...model (using) G-banding patterns or the distribution of structural proteins along the chromosome.”

(191-197) Auxin-inducible depletion of TOP2A resulted in cells with hypo-condensed chromosomes. There were only minor changes in stiffening behaviour. These hypo-condensed chromosomes had altered H2B-eGFP-labelling. Interestingly, the brighter areas extended less than the dimmer regions. Perhaps this gives a clue to the scale of the chains? This should be reviewed in the Discussion.

We agree with the reviewer that figure 5 was not sufficiently informative and appreciate their suggestion for the inclusion of an improved figure. We have now replaced figure 5 with a larger and more descriptive cartoon in extended data figure 3 depicting how the HWLC model conceptually captures chromosome mechanics. Specifically, this model describes how two levels of heterogeneity in chromosome structure might impact its nonlinear mechanical response: 1) (top row) spatial heterogeneity, where different spatially separated regions exhibit distinct mechanical properties, leading to heterogenous deformations. 2) (bottom row) scale-dependent heterogeneity, where chromosomal structure on different length scales has distinct mechanical properties reflecting differences in chromosome organization at different scales. In the HWLC model, these levels of heterogeneity are captured by combining WLCs with distinct mechanical properties in a hierarchical fashion. We believe that the new figure conveys the key idea that we propose with this model: as a function of force different spatially and hierarchically separated components of the chromosome dominate the mechanical response of the chromosome as a whole. Notably, this heterogeneity is not described by a single length scale, but processes at many different length scales might be involved. Therefore, while the different mechanical properties observed for denser/brighter and less dense/dimmer regions of hypo-condensed chromosomes agree with the HWLC model, their size may reflect just one of many length scales involved. In the manuscript, we show how this conceptual coarse-grained HWLC model quantitatively describes the nonlinear mechanical response of mitotic chromosomes. Of course, a future challenge will be to show how properties of the HWLC model that we fitted to the data link to specific molecular components of chromosome architecture.

If possible, incorporation of G banding data (or other underlying structural information) into the study would be of broad interest and would strengthen the MS.

We agree with the referee that this would be ideal. However, to our knowledge, the only well-established methodology for analysis of chromosome domains on chromosomes is via G-banding. Unfortunately, this utilizes a protocol is not compatible with our analysis of native chromosomes in solution for two reasons. First, G-banding requires that the chromosomes be fixed and dried, which drastically alters the overall structure. Second, it requires digestion of chromatin proteins with trypsin or a similar protease. This is not compatible with our system because chromosome handling in the tweezers depends on the use of protein-mediated handles on the chromosome surface. Moreover, the structure of the chromosome itself would inevitably be affected by treatment with a protease and Giemsa.

2. In the manuscript it is stated that “through inducible degradation of TOP2A in cells with already condensed mitotic chromosomes, we provide support for a role of TOP2A in the maintenance of a

compacted chromosome structure.” However, the protocol as outlined in Ext. Data Fig. 4 would result in TOP2A levels decreasing as cells enter M phase, rather than being degraded from chromosomes already in prometa/metaphase. Given that cells released from the arrest will not move into M phase completely synchronously in some cells, chromosomes may already have undergone extensive compaction before TOP2A levels fall, while others will be compacting chromosomes where the protein has been largely removed. Therefore, the population examined will be very heterogeneous, which may explain the two populations of TOP2A-depleted chromosomes the authors describe: a hypocondensed population, which may reflect cells where compaction has occurred in the absence of TOP2A (given that lack of TOP2A has been shown by others to result in a failure of condensin to compact chromosomes beyond a thin ribbon-like state, however long they remain arrested in nocodazole), while those with a more normal appearance may be from cells in which chromosome compaction, through the action of both condensin I and II, had taken place prior to TOP2A degradation.

The effect of TOP2A removal on the maintenance of metaphase chromosomes would be better discerned by releasing cells from G2/M arrest into nocodazole 1 hr before the addition of auxin, with chromosomes then sampled at ~2 hr +/- auxin. In addition, to control for the fact that, in cells expressing TOP2A, chromosomes will continue to compact while arrested in nocodazole, the structure of chromosomes from which TOP2A has been degraded should be compared with TOP2A-containing chromosomes sampled at the point when auxin was added into the culture (rather than with controls sampled after further arrest in nocodazole).

We thank the referee for these comments. There are two issues being raised here. First, that the cells will release into mitosis in an unsynchronized manner and, second, that TOP2A will be degraded while chromosome condensation is occurring and not only when the chromosomes have already undergone condensation in prometaphase. For the first issue, the cells emerge from G2 arrest in a remarkably synchronous way. Indeed, this is the reason that the CDK1^{AS} synchronization system was adopted for this study because the release kinetics are so rapid and uniform in the cell population. We present these new data in a modified extended data figure 5. For the second issue, the kinetics of auxin-induced degradation in mitosis are such that the level of TOP2A will not be strongly decreased before the cells arrive in prometaphase (which occurs as soon as 15 mins after release in ~80% of the cells using the CDK1^{AS} system, extended data figure 5)). Indeed, 90% of cells are in prometaphase 25 mins after release. We find that TOP2A depletion in mitotic cells (as opposed to degradation in an asynchronously growing cell population as was reported previously) currently takes more than 1 hour to proceed to the point where the protein is >75% depleted, and therefore the vast majority of the effect that we see following the degradation of TOP2A does in fact occur on chromosomes that are already compacted. Moreover, even when cells are arrested in Nocodazole for several hours before TOP2A is depleted, the resulting chromosome population is just as heterogeneous as it is when TOP2A begins to be depleted during early mitosis (Nielsen et al. 2020, PNAS, Fig. S3C,E). Nevertheless, we fully accept that the phenotype we analyze is not a pure chromosome maintenance phenotype and have amended the text accordingly.

3. In this reviewer's view, it is uncertain whether the data from the swelling and re-compaction expts (Fig. 4) can be used to argue for TOP2A providing a structural memory. Since TOP2A is a major non-histone protein component of the mitotic chromosome, the obvious criticism is that its removal may simply give rise to perturbation of structure arising from extensive depletion of protein making up the

chromosome, rather than reflecting a specific role of TOP2A (for example, a role as an internal linker, as suggested for condensin). One way to address this might be to examine the impact of catalytically inactive forms of TOP2A. While expressing an enzymatically-dead form of the enzyme in living cells may not be feasible (because of dominant-negative effects), this could be looked at through incubation of isolated chromosomes with purified protein prior to structural analysis. If the effects on structure, detected when chromosomes have been prepared from auxin-treated cells, can be restored by incubating isolated chromosomes with purified human WT TOP2A (-/+ ATP or with a nonhydrolyzable ATP analogue), this rescue could be compared to that seen when using other versions of TOP2, such as TOP2B, or mutant forms of TOP2A (e.g. catalytically-dead Y805S, or catalytically-perturbed forms such as K662A, or a version deleted for the C-terminal Chromatin Tether Domain, that has been shown to be required for TOP2A to associate robustly with mitotic chromatin). These expts would help to determine whether it is the presence of catalytically-active TOP2A that is required, and whether other activities of this protein contribute to metaphase structure.

We thank the reviewer for the very helpful suggestions. As discussed above in the response to Referee #1, we acknowledge that the term “structural memory” is too sweeping to be supported by our data.

With regard to the helpful suggestion of adding back purified WT or catalytically dead TOP2A protein to purified chromosomes, we have attempted to confirm that we can observe robust association of exogenous protein with chromosomes (using fluorescently labeled protein), but this was unsuccessful. With hindsight, given that TOP2A association with mitotic chromosomes in cells is tightly regulated by SUMOylation and other post-translational modifications, and the fact that it is loaded to chromosomes prior to full condensation, it is probably not surprising that recombinant TOP2A in solution is unable to replace the TOP2A removed from condensed chromosomes following auxin treatment of cells. Moreover, we consider it highly unlikely that the effect we see on chromosome structure/compliance following TOP2A depletion simply reflects a loss of protein in general from the chromosome. This is because depletion of another major structural protein - Condensin I - has dramatically different effects on the structure and shape of metaphase chromosomes from that seen when depleting TOP2A (or indeed from that seen when depleting Condensin II). Unfortunately, therefore, this idea could not be directly tested and hence we rephrased this section of the text and toned down our conclusions.

Minor comment:

Some of the terms used within the MS are not widely used in biology labs, and a brief explanation within the text might save readers time checking definitions. These include: storage modulus, loss modulus, stiffening exponent.

We apologize for this omission and have now defined these terms on the first occasion that they are used in the text.

Reviewer Reports on the First Revision:

Referee #1:

The authors have responded well to the reviews, and we recommend publication.

Referee #2:

The authors have dealt with each of the points raised to my satisfaction, and I believe that this work is now suitable for publication in the journal.